# G-quadruplex DNA structures in human stem cells and differentiation

Katherine G. Zyner [1,4], Angela Simeone[1,4], Sean M. Flynn [1], Colm Doyle[1], Giovanni Marsico[1], Santosh Adhikari [2], Guillem Portella[2], David Tannahill [1] & Shankar Balasubramanian [1,2,3 ✉]

The establishment of cell identity during embryonic development involves the activation of specific gene expression programmes and is underpinned by epigenetic factors including DNA methylation and histone post-translational modifications. G-quadruplexes are four-stranded DNA secondary structures (G4s) that have been implicated in transcriptional regulation and cancer. Here, we show that G4s are key genomic structural features linked to cellular differentiation. We find that G4s are highly abundant in human embryonic stem cells and are lost during lineage specification. G4s are prevalent in enhancers and promoters. G4s that are found in common between embryonic and downstream lineages are tightly linked to transcriptional stabilisation of genes involved in essential cellular functions as well as transitions in the histone post-translational modification landscape. Furthermore, the application of small molecules that stabilise G4s causes a delay in stem cell differentiation, keeping cells in a more pluripotent-like state. Collectively, our data highlight G4s as important epigenetic features that are coupled to stem cell pluripotency and differentiation.

[1] Cancer Research UK Cambridge Institute, Li Ka Shing Centre, Robinson Way, Cambridge CB2 0RE, UK. [2] Yusuf Hamied Department of Chemistry, University of Cambridge, Lensfield Road, Cambridge CB2 1EW, UK. [3] School of Clinical Medicine, University of Cambridge, Cambridge CB2 0SP, UK. [4] These authors contributed equally: Katherine G. Zyner, Angela Simeone. ✉email: sb10031@cam.ac.uk

Human embryonic stem cells (hESC) can indefinitely self-renew yet retain the capacity to generate all embryonic cell lineages[1]. hESC pluripotency is governed by a core network of master transcription factors (TFs), including OCT4, NANOG and SOX2[2], and their interplay with signalling pathways as well as chromatin and histone modifiers. This hESC TF network sustains the pluripotent state by simultaneously promoting chromatin plasticity whilst repressing key genes involved in lineage commitment through the establishment of unique epigenetic architecture. More specifically, the hESC epigenome comprises decondensed chromatin, high levels of active histone modifications and low levels of heterochromatic proteins[3,4]. Promoters primed for activation upon differentiation can also be marked by both active (H3K4me3) and repressive (H3K27me3) histone marks, allowing genes to be up or down-regulated in response to the appropriate developmental stimulus[5]. During differentiation, the epigenetic landscape is reorganised through increased DNA methylation, decreased chromatin accessibility and histone mark reassignments. Such changes during differentiation lead to a reorganisation of global 3D chromatin architecture to enable the activation of lineage-specific gene expression programmes and silencing of unrelated programmes[6–8]. This is underpinned by rearrangements in promoter-enhancer interactions and super-enhancers; large clusters of regulatory elements that serve as docking sites for TFs, epigenetic modifiers and basal transcriptional machinery[6–8].

Of emerging importance to genome architecture and function are alternative DNA secondary structures known as G-quadruplexes (G4s) that form from certain guanine-rich sequences[9] (Fig. 1a). G4s arise from the self-association of guanine bases through Hoogsteen hydrogen bond base pairing (Fig. 1a). Computational analyses show that nucleic acid sequences with G4-forming potential are evolutionarily conserved[10] and non-randomly dispersed across the genome being particularly enriched in promoters, recombination sites and telomeres[11,12]. Experiments using an antibody, specifically raised against G4 secondary structures, reveal that the endogenous G4 genome landscape is tightly regulated since only ~1–2% of over 700,000 human sequences can biophysically fold into a G4 structure in vitro[13], actually do so within a chromatin context[14,15]. Previously, we showed that G4s have an increased prevalence in cancer cells[16] and are particularly associated with highly expressed and amplified genes in patient-derived aggressive breast cancer tissue[17]. In keeping with computational predictions, endogenous G4s in chromatin are primarily located in nucleosome-depleted regions (NDRs) of highly active promoters[14,18,19]. Accumulating evidence suggests that G4s act as recruitment hubs for TFs[15,20,21], are involved in interactions[21] between distal genomic loci and can interact with/ modulate DNA methyltransferases[22,23], which further connects G4s to transcriptional and epigenome regulation[9,24]. Given the extensive evidence, particularly from cancer cell studies, that G4s modulate the functions of the (epi)genome, we now present data on the potential roles of G4s in human embryonic stem cells and their differentiation into downstream lineage-specific states.

## Results

**G4 abundance in pluripotent human embryonic stem cells is lost during differentiation**. To gain insights into the role of G4s during development, we utilised a human embryonic stem cell (hESC) system which enables differentiation to be studied in vitro and is an accepted model that recapitulates many aspects of cell lineage specification during human embryogenesis[25]. hESCs were differentiated into two well-characterised, multipotent stem cells each with differing lineage potentials: cranial neural crest cells (CNCCs)[26] and neural stem cells (NSCs[27], Fig. 1b, c). NSCs undergo self-renewal and can generate neurons, astrocytes, and oligodendrocytes of the central nervous system[28]. CNCCs also self-renew but have a broader lineage potential giving rise to cranial neurons and glia as well as craniofacial mesodermal derivatives, such as bone, cartilage and smooth muscle[29,30]. We first confirmed stem cell derivation and identity using a range of established cell-lineage-specific antibody markers by immunofluorescence microscopy (IF), flow cytometry experiments and RNA-seq (Fig. 1c and Supplementary Fig. 1). For example, hESCs showed specific expression of OCT4 (POU5F1) and NANOG, NSCs specifically expressed PAX6 and SOX1 and CNCCs specifically expressed TFAP2A and TWIST1.

To capture the incidence of G4s in cellular chromatin we deployed our well-characterised G4 structure-specific antibody, BG4[31] in chromatin immunoprecipitation sequencing (G4-ChIP-seq)[14] experiments. Hereafter, we define G4s as G4-ChIP-seq-positive regions identified in at least 2 out of 3 biological replicates (see Methods section, Supplementary Fig. 2). We found 17950 robust G4s in pluripotent hESCs and upon their differentiation into CNCCSs or NSCs we observed 9456 and 4436 respectively (Fig. 1d). We confirmed that BG4 was indeed recognising a broad spectrum of G4 structural types (Supplementary Fig. 2h, i) as for previous studies[14,32]. Corroborating this finding, G4 loss upon differentiation was observed at the single-cell level via BG4 immunofluorescence (IF) staining (Supplementary Fig. 3a, b). G4 loss was also primarily independent of the cell cycle stage (Supplementary Fig. 3c–e and Supplementary Discussion).

**Most G4s in differentiated stem cell lineages are present in pluripotent embryonic stem cells**. Remarkably, over 75% of G4s found in CNCCs or NSCs were also found in hESCs (Fig. 1e, f and Supplementary Fig. 4a, b). Despite the distinct lineages of NSCs and CNCCs, 83% (3,694/4,436) of NSC G4s were also observed in CNCCs (Fig. 1f and Supplementary Fig. 4c) and 77% (3426/4436) of NSC G4s were common to both CNCCs and hESCs (Supplementary Fig. 4d). Therefore, while G4s are lost during differentiation, G4s present in differentiated daughter cells largely occur at genomic locations where G4s were already present in the hESC state.

When hESCs differentiate, chromatin accessibility generally decreases as cell lineages become specified[7,8], which we confirmed by Assay for Transposase-Accessible Chromatin (ATAC-seq, Supplementary Fig. 5a). Over 99% of G4s are located at NDRs regardless of stem cell type (Supplementary Fig. 5b) and is similar to our previous observations in cancer[14]. Controlled differentiation provides the opportunity to evaluate whether G4s and chromatin structures are tightly coupled during cell state transitions. We, therefore, compared common and stem cell-type-specific G4s (Supplementary Fig. 5c, d) and open chromatin sites (as identified by ATAC-seq, Supplementary Fig. 5e, f) for hESCs, CNCCs and NSCs. Upon differentiation, we observed that G4 loss is accompanied by a corresponding loss of ATAC signal, conversely, the establishment of new G4s in differentiated daughter cells leads to a corresponding gain in ATAC signal. Hence, G4s in human stem cells are intimately linked to chromatin accessibility.

**G4s are positioned within stem cell regulatory regions**. To investigate possible functions of G4s in human stem cells, we next explored the locations of G4s in each stem cell type (Supplementary Fig. 6a). For each stem cell type, G4s were found primarily in promoters, enhancers, super-enhancers and sites occupied by lineage-specific transcription factors, which are of

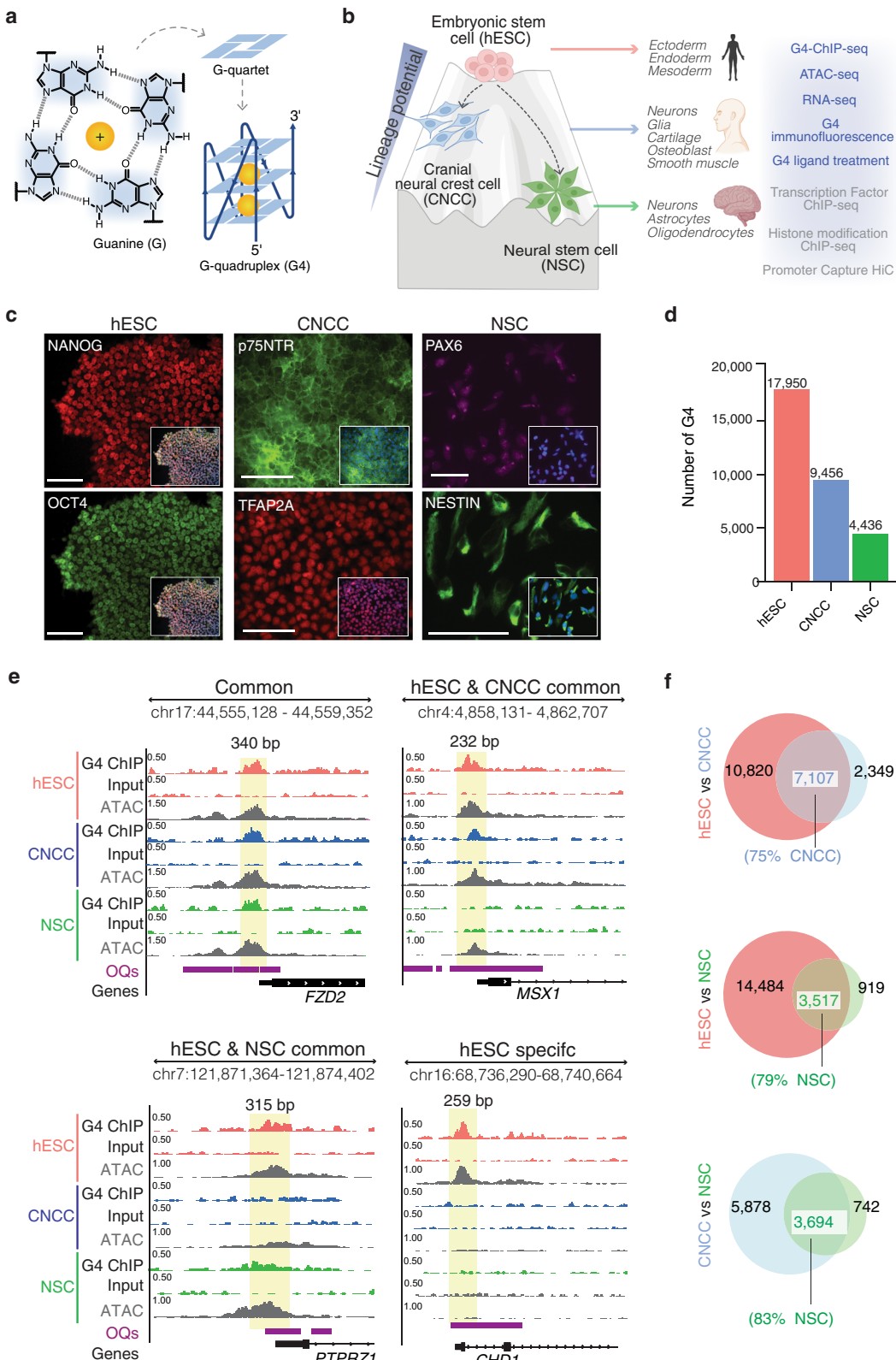

vital importance to transcriptional programmes and cell identity[33–35] (Fig. 2 and Supplementary Fig. 6). This included the binding sites of important core pluripotency master regulators OCT4, NANOG and KLF4 in hESCs[2,36], the neural crest master regulators TFAP2A and NR2F1 in CNCCs[37], the neuroectoderm lineage specification transcription factor PAX6 in NSCs[38] (Fig. 2c) and the H3 histone acetyltransferase p300, involved in

enhancer activity[39] (Supplementary Fig. 6c). Notably, cell-type-specific G4s were found to be enriched at cell-type-specific enhancers while G4s common to two or more stem cell types were more enriched in promoter regions (Fig. 2a). Comparing our G4s maps with promoter-capture Hi–C maps for hESCs[40] and hESC-derived neuronal progenitor cells (developmentally analogous to NSC[41]), revealed that G4s are also enriched in both

**Fig. 1 G4 abundance is linked to degree of stem cell plasticity. a** G-quartet with 4 Hoogsteen-base-paired guanines, which stack to form a G-quadruplex (G4) structure stabilised by cations (e.g., $K^+$). **b** Overall study design and schematic showing differing lineage potential of stem cell types generated and analysed. Data generated in this study are indicated in blue text, with published datasets indicated in grey text. Schematic created with BioRender.com. **c** Immunofluorescent microscopy images for hESC (NANOG, OCT4), CNCC (p75NTR and TFAP2A) and NSC (PAX6 and NESTIN) markers. Inset: cell lineage marker merge with DAPI. Scale bar = 50 µm. **d** Number of confirmed G4-ChIP-seq sites (defined henceforth as "G4s") identified in each stem cell type. **e** Genome browser view of G4 signal for all three stem cell types across the promoters of the genes *FZD2, MSX1, PTPRZ1* and *CHD1*. Yellow box highlights regions where G4s overlaps open chromatin (defined by ATAC-seq; ATAC) and genome sites that have the ability to fold into G4 structures in vitro (called OQs, observed quadruplex sequences[13]). Genomic coordinates are indicated at the top. **f** Venn diagrams illustrating overlap of G4s between hESCs and CNCCs (top) or hESC and NSCs (middle) or CNCCs and NSCs (bottom).

promoters and promoter-interacting regions (Fig. 2e). G4s are also prevalent at the binding sites of known architectural proteins involved in promoter-enhancer chromosomal loop anchorage and formation in hESC (Fig. 2f). These observations implicate G4s in the long-distance regulation of promoter-enhancer interactions and overall 3D structural organisation of the genome[42] in stem cells. The majority of G4s were also located at hypomethylated CpG islands (> 60% of all G4s) in hESCs (Supplementary Fig. 6d) and is consistent with our earlier findings that showed an inverse relationship between G4 signals and proximal cytosine methylation in cancer cells[22]. Overall, our findings highlight an intimate relationship between the incidence of folded G4s in genomic regulatory regions and stem cell identity.

**G4s in promoters are associated with H3K4me3 and bivalent promoter transitions during differentiation**. In addition to promoters exclusively marked by either activating (H3K4me3) or repressive (H3K27me3) histone marks, bivalent promoters are classified by having both H3K4me3 and H3K27me3 (Fig. 3a). Bivalent promoters are prevalent in stem cells and are particularly important for developmental and lineage-specific genes that are generally repressed or lowly expressed, but are poised for rapid activation upon differentiation[5]. We next examined the relationship of folded G4s to different promoter classes and transcription. Most promoters with a G4 in each stem cell type were also marked by H3K4me3 (67–94%, Fig. 3b). G4s were also found to be enriched in bivalent promoters (Fig. 3c, d and Supplementary Fig. 7a, b) with almost half (3041/6350) of all hESC bivalent promoters carrying a G4 (Supplementary Fig. 7c). This relationship is also apparent from the coincidence of hESC G4s at the binding sites of RING1B (Fig. 2f), a PRC1 component generally found at bivalent domains[43]. Our observation shows that folded G4 structures can physically co-exist with repressive histone marks in the context of bivalency. This provides experimental support for findings from computational studies that predict G4 sequence motif association with bivalent and polycomb-associated chromatin at DNA[44–46]. Our data thus highlight G4s as prevalent structural features in poised genes involved in developmental decisions.

During lineage specification, bivalent promoters generally become transcriptionally repressed or active, concomitant with loss of either H3K4me3 or H3K27me3, respectively[8,47]. There are four combinations of promoter G4 transitions that are characterised by G4 presence (G4+) or absence (G4−) in the initial state (hESCs; $G4_E$) and daughter cells ($G4_D$) for a given gene (Supplementary Fig. 8a, Methods section). Hereafter we refer to "G4 maintenance" to describe when promoter G4s are present in both hESCs and the differentiated daughter cell. Significantly more hESC bivalent promoters ($p < 3.0E-05$) transition to an active H3K4me3 status in CNCCs when maintaining ($G4_E+$ $G4_D+$, 79%) or gaining ($G4_E−$ $G4_D+$; 83%) a G4, as compared to those that lose ($G4_E+$ $G4_D−$; 40%) or never had a G4 ($G4_E−$ $G4_D−$, 30%, Fig. 3e). Conversely, hESC bivalent promoters that either lose or do not gain a G4 are more likely to

($p < 2.6E-09$) maintain bivalency $G4_E+$ $G4_D−$; 43%; $G4_E−$ $G4_D−$; 43%) or become repressed (H3K27me3; $G4_E+$ $G4_D−$, 10%; $G4_E−$ $G4_D−$; 18%) after differentiation (Fig. 3e). Thus, maintenance or acquisition of a promoter G4 appears to be related to the transition of bivalent hESC promoters towards an active H3K4me3 state in CNCC differentiation. Such bivalent promoters were also found to have on average higher H3K4me3 levels (Fig. 3f and Supplementary Fig. 7d–f), chromatin accessibility (Fig. 3g and Supplementary Fig. 7g, h) and gene expression (Fig. 3h, Supplementary Fig. 9 and Supplementary Discussion) compared to bivalent promoters without a G4. Furthermore, hESC bivalent promoters which maintain a G4 but lose repressive H3K27me3 in CNCC differentiation are strongly associated with developmental signalling pathways such as WNT signalling[29] (Supplementary Fig. 8b and Supplementary Data 1). In hESCs, the other main G4-positive promoter type is marked by H3K4me3. Significantly more ($p < 1.4E-07$) of these promoters retain their H3K4me3 status on CNCC differentiation when maintaining ($G4_E+$ $G4_D+$; 98%) or gaining ($G4_E−$ $G4_D+$; 99%) a G4, as compared to those that either lose ($G4_E+$ $G4_D−$; 79%) or never had a G4 ($G4_E−$ $G4_D−$; 84%) (Fig. 3e). The majority (>75%) of H3K4me3 and bivalent G4-positive promoters in CNCCs also appear to have arisen from promoters in hESCs that also carried a G4 (Supplementary Fig. 8c). Therefore, G4 maintenance during differentiation appears to be linked with propagation of the hESC histone promoter status to the daughter cells. The same G4-associated histone-mark transitions were also seen in the differentiation of hESCs into NSCs (Supplementary Fig. 8d–g), suggesting that the coupling of G4s with histone marks is a common feature of genes involved in transitioning to a more differentiated state.

**Maintenance of promoter G4s is linked to transcriptional stabilisation between stem cell transitions**. Many genes undergo transcriptional changes upon hESC differentiation into downstream cell lineages[6,8]. To understand how promoter G4 status relates to changes in gene expression during differentiation, the transcript levels in hESCs were compared to those in each daughter cell for the four combinations of promoter G4 transitions (Supplementary Fig. 8a). We used a weighted linear regression model to fit expression level data and computed the residuals (i.e., distance from the regression line) to quantify transcriptional variability between the two cell states being considered (see Methods secion). $G4_E+$ $G4_D+$ promoters showed significantly lower ($p < 2E-275$) transcriptional variability between hESCs and daughter cells than $G4_E−$ $G4_D−$, $G4_E+$ $G4_D−$ or $G4_E−$ $G4_D+$ promoter transitions (Fig. 4a–c and Supplementary Figs. 10a, b, 11, 12 and 13). This stabilisation is also observed when considering differential gene expression (FDR < 0.05, abs (Log2FC) >1) (Supplementary Fig. 10c, d): the proportion of non-differentially expressed (DE) genes for $G4_E+$ $G4_D+$ promoters is ~ 2–3-fold higher ($p < 2E-312$) than genes that are DE. In contrast, for $G4_E−$ $G4_D+$ and $G4_E+$ $G4_D−$ promoter transitions the proportion of DE genes is significantly

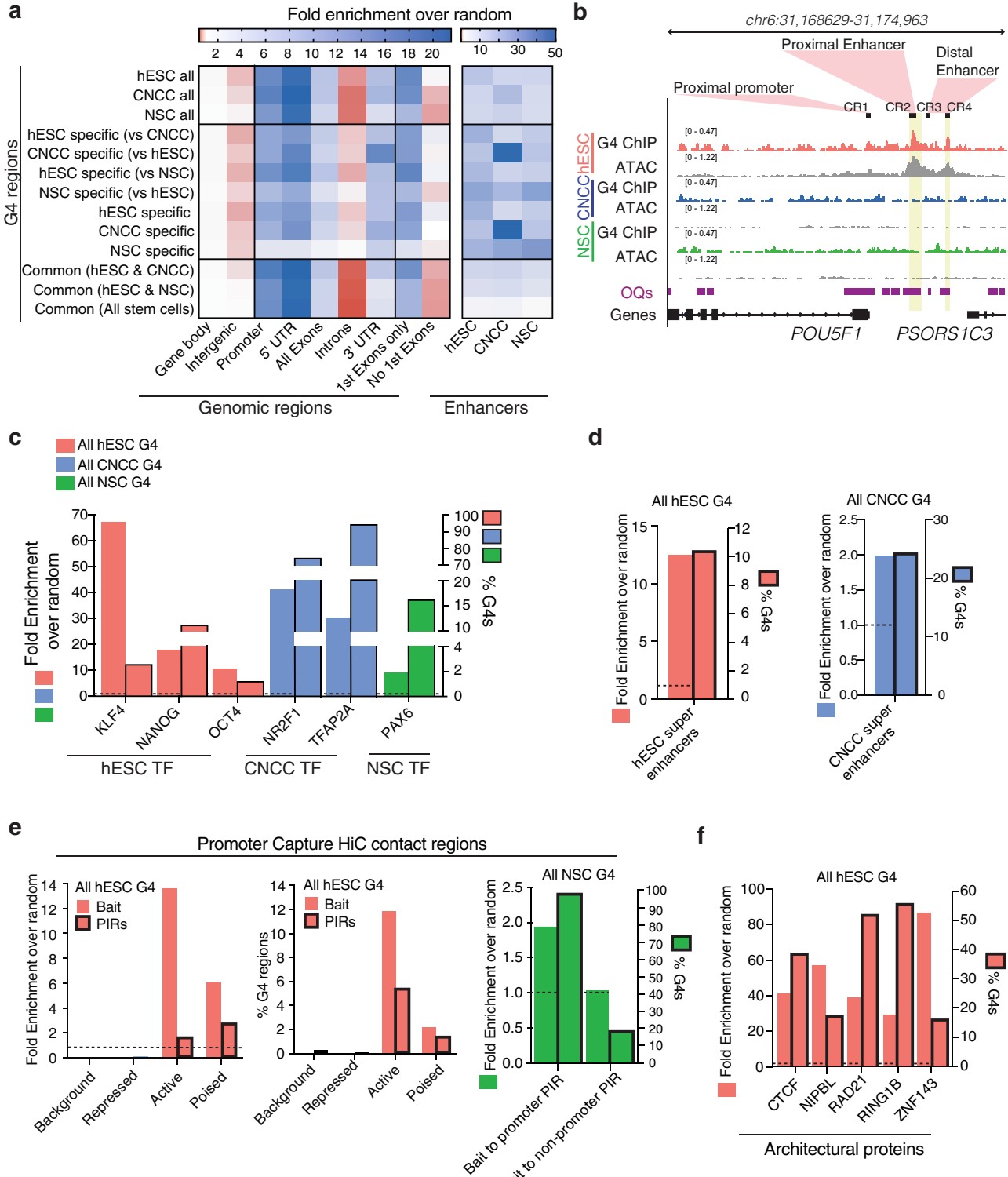

**Fig. 2 G4s are found in stem cell regulatory elements. a** Fold enrichment over random ($n = 1000$ permutations) for G4s at genomic features from the reference human annotation GENECODE v.28 or at enhancers, defined as promoter distal H3K27ac. UTR untranslated region; Promoter defined as ±1 kb around transcription start site. **b** Genome browser view of G4 signal across hESC-specific proximal and distal enhancer of *POU5F1* (as defined in Yang et al.[89]). CR conserved region. The yellow box highlights regions where G4s overlaps open chromatin (ATAC) and genome sites which have the ability to fold into G4 structures in vitro (OQs[13]). **c-f** Fold-enrichments over random ($n = 1000$ permutations) and proportion of G4s per stem cell type at **c** the binding sites of cell-specific transcription factors (TF); **d** super-enhancer elements as defined in Hnisz et al.[34] and Wilderman et al.[90]; **e** promoter (bait) and promoter-interacting regions (PIRs) from promoter-capture HiC experiments (as defined in Freire-Pritchett et al.[40]). Centre and left panel: hESCs active promoters (H3K4me3 and/or H3K27ac), poised promoters (H3K4me3 and H3K27me3), repressed promoters (H3K27me3) and background (none). Right-most panel: NSC promoter-promoter PIRs and promoter-non-promoter PIR interactions of developmentally analogous H1-hESC-derived neural progenitor cells from Jung et al.[41]; **f** binding sites of chromatin architecture proteins involved in promoter-enhancer looping.

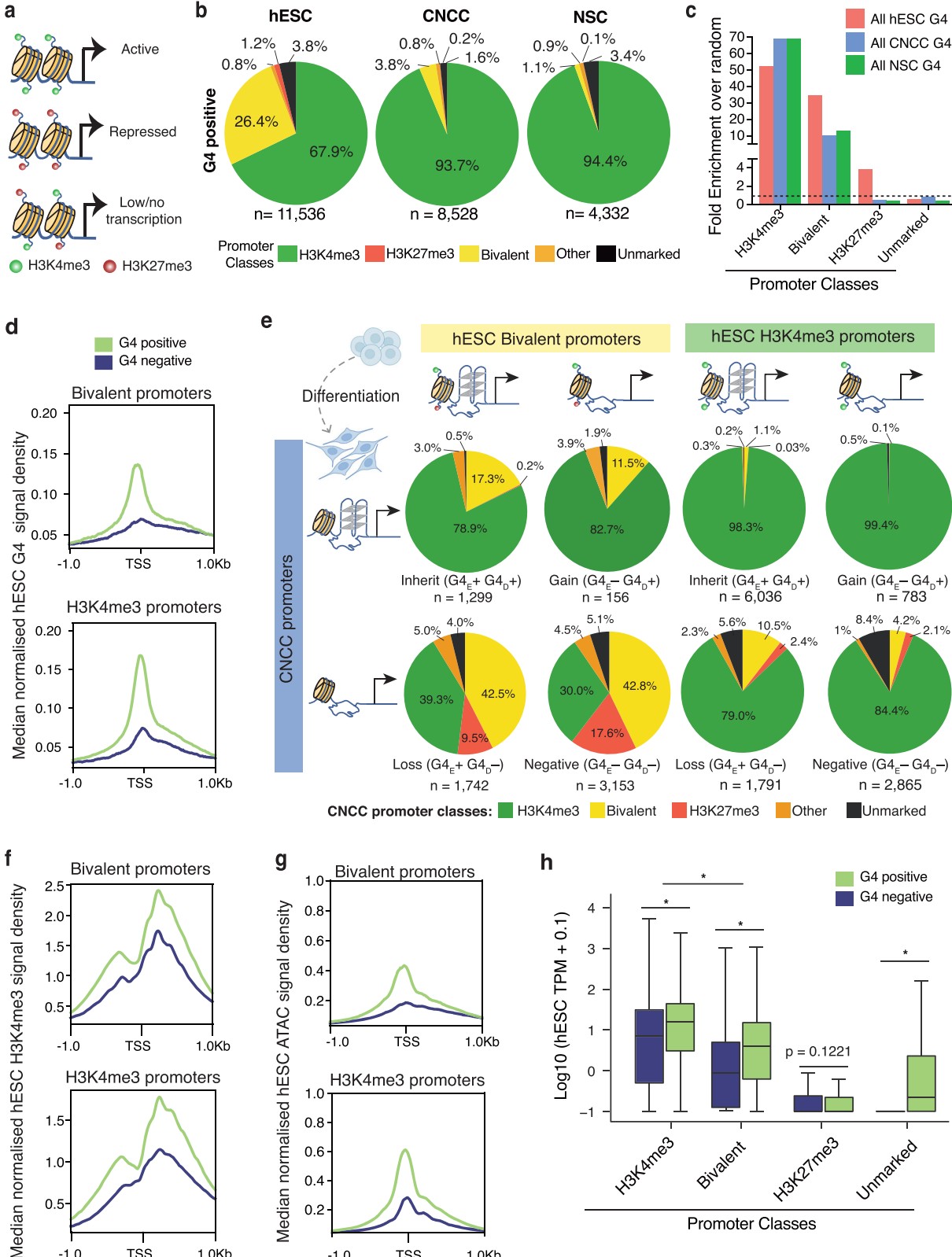

($p < $8E-14) higher than that of non-DE genes. This reveals that genes that lose, acquire, or never had a promoter G4 tend to have greater transcriptional changes during differentiation.

Promoter-related genetic and epigenetic features[48], including H3K4me3[49], are known to play a role in minimising transcriptional variability within a cellular population. This is thought to ensure the consistent expression of key genes when confronted with a fluctuating environment e.g during cellular differentiation. While we also observed transcriptional stabilisation for genes that maintain H3K4me3[49] ($R^2 = 0.75$ (CNCCs), 0.70 (NSCs)) and chromatin accessibility ($R^2 = 0.75$ (CNCCs), 0.74 (NSCs)) surprisingly the effect observed for G4s was markedly stronger ($R^2 = 0.84$ (CNCCs), 0.85 (NSCs); Supplementary Fig. 13c, d). Gene Ontology (GO) enrichment analysis for genes that are not

**Fig. 3 G4 presence in promoters is associated with bivalent and H3K4me3 promoter class transitions in differentiation. a** Schematic of H3K4me3, H3K27me3 and bivalent promoters and their associated transcriptional activity in stem cells. **b** Pie charts showing the proportion of promoters containing a G4 (G4 positive) in each promoter histone modification class: H3K4me3 (enrichment of H3K4me3 only), H3K27me3 (enrichment of H3K27me3 only), bivalent (overlap of H3K4me3 and H3K27me3 enrichments), other (non-overlapping H3K4me3 and H3K27me3 enrichments) and unmarked (negative for H3K4me3 and H3K27me3 enrichments). **c** Fold enrichment over random ($n = 1000$ permutations) of G4s in the promoter histone modification classes defined in **b** for each stem cell type. **d** Median normalised ChIP-seq hESC G4 signal density at hESC bivalent (top) and H3K4me3 (bottom) promoter regions. TSS transcription start site. **e** Distribution of histone modification transitions for bivalent and H3K4me3 hESC promoters after differentiation to CNCCs, segregated by G4 promoter types (Supplementary Fig. 9a, see Methods section): $G4_E + G4_D +$: G4 maintenance from hESCs to daughter cells; $G4_E + G4_D -$: G4 present in hESCs and lost in daughter cells; $G4_E - G4_D +$: G4 gained in daughter cell and $G4_E - G4_D -$: promoters lacking a G4 in both hESCs and daughter cells. **f, g** Median normalised H3K4me3 read density (**f**) and ATAC-seq signal (**g**) across hESC promoter subtypes as described in **d**. **h** Gene expression levels (log10[average TPM + 0.1], transcript per million averaged across biological replicates) for hESCs. Histone modification promoter classes as defined in **b**. Data are presented as median (centre) and interquartile range (box; the lower and upper bounds of the box represent the 25th and 75th percentiles, respectively). Whiskers represent ±1.5x interquartile range. $N = 5$, 3 and 4 biologically independent samples for hESCs, CNCCs and NSCs, respectively. Number of genes per group: ESC G4 positive (H3K4me3: 7827, bivalent: 3041, H3K27me3: 137 and unmarked: 400) and ESC G4 negative (H3K4me3: 3648, bivalent: 3309, H3K27me3: 1390 and unmarked: 38491). *$p < 2E{-}16$, one-sided Kolmogorov–Smirnov test.

differentially expressed and were $G4_E + G4_D +$ upon differentiation, revealed essential cellular programmes as highest-ranking terms (FDR < 0.05) including splicing (e.g. *SNRNP70, HNRNPA2B1, SF3B1)*, cell cycle (e.g. *CUL3, PML*), metabolism (e.g. *VCP, PHF23*), translation, ubiquitination (e.g. *DERL3, CUL3*) and chromatin maintenance (e.g. *KANSL2, KDM1A*; Fig. 4d, e). For selected genes, genome browser views and validation of G4 presence and gene expression by ChIP-PCR and RT-PCR respectively are presented in Supplementary Figs. 14 and 15). There was substantially lower enrichment in these pathways for genes with a $G4_E - G4_D -$ promoter signature. KEGG, Reactome and Wikipathway functional analyses also confirmed these findings (Supplementary Data 2 and 3). Overall, these results show that G4 maintenance may be an important chromatin feature linked to the transcriptional stabilisation of genes essential for key cellular functions.

**Promoter G4 landscape changes are linked to stem cell identity**. Genes that acquire a promoter G4 ($G4_E - G4_D +$) during differentiation generally have increased expression in the daughter cell compared to hESCs (Supplementary Fig. 10c, d). For CNCCs, the highest-ranking significant GO terms for this category were related to cell specification and include the key CNCC regulators *TWIST1, TWIST2* and *SNAI2*[29] (Fig. 5a, b and Supplementary Figs. 15 and 16a–e). Likewise, for NSCs, many genes with increased expression that acquire a G4 were related to neurodevelopmental pathways such as Notch signalling e.g. *DLL1* and *HES1* essential for NSC maintenance[50], *SOX11* essential for NSC fate specification[51] and *RFX4*, which when disrupted leads to neurodevelopmental disease[52,53] (Fig. 5c, d and Supplementary Figs. 15 and 16a–e). Acquisition of G4s during differentiation may therefore play a role in lineage specification. Conversely, genes that lose a promoter G4 ($G4_E + G4_D -$) upon differentiation are generally down-regulated in the daughter cell (Supplementary Fig. 10c, d). For this latter category, top-ranking GO terms include genes associated with cellular tight junctions (e.g. hESC-specific *CHD1*[54]) and alternative developmental lineages (e.g. GO: blood circulation and GO: circulatory system processes), and genes (e.g. *PRDM14*[55] and *TERT*[56]) and signalling pathways (MAPK, RAP1, RAS and PI3K-AKT) important for hESC pluripotency and self-renewal[57] (Supplementary Figs. 15 and 16f–h). Thus, some G4s only exist in the hESC cell state and are specifically linked to the expression of key pluripotency genes whose expression is subsequently lost upon differentiation.

**G4 stabilisation with small molecules delays hESC differentiation**. For differentiation to proceed, the pluripotency

transcriptional network that maintains self-renewal must be dismantled and lineage-specific transcriptional programmes activated[58]. G4s frequently occur in stem cell regulatory regions with many G4s being lost during differentiation, concomitant with the silencing of their associated genes. If G4s are coupled to gene expression, then introducing a factor that preserves G4 as folded structures in hESCs might present a barrier to differentiation. We tested this hypothesis by adding a small molecule that stabilises G4 structures to hESCs. OCT4-EGFP-expressing hESCs were differentiated into CNCCs[59] in the presence or absence of different concentrations of PhenDC3, a small molecule selective for G4s[60] (Fig. 6a), which does not result in detectable DNA damage or cell cycle check point arrest upon cellular treatment (Supplementary Fig. 17a–e). Exit from pluripotency was monitored by measuring the loss of OCT4-EFGP by flow cytometry and IF, while CNCC differentiation was determined by IF microscopy for SOX10, a definitive marker of CNCCs[29] (Fig. 6b, c and Supplementary Fig. 17f, g). Loss of OCT4 (~50 and 90% at day 3 and 5 respectively) and gain of SOX10 (~5 and 40% at day 3 and 5 respectively) expression was first confirmed during differentiation in DMSO controls (Fig. 6b–d). From day 3 of differentiation, there was a PhenDC3 dose-dependent significant ($p < 0.01$) increase in the proportion of live cells (6-37%) expressing OCT4-EGFP compared to DMSO-treated controls (Supplementary Fig. 17f, g). This result was confirmed by IF analyses: PhenDC3 treatment resulted in 2- and 4-fold more OCT4 expressing cells on day 3 and 5, respectively (Fig. 6b, c), which was accompanied by up to a 4-fold decrease in the induction of SOX10-expressing cells (~10%) compared to DMSO controls (41%) (Fig. 6c, d). Global transcriptome analysis by RNA-seq verified that differentiating hESCs cultured in PhenDC3 had a transcriptional signature more similar to undifferentiated hESCs than to the differentiated controls (Fig. 6e and Supplementary Fig. 18). Compared to DMSO and non-treated controls, treatment with PhenDC3 lead to fewer genes being differentially expressed during differentiation (37% less at day 3 and 43% less at day 5). Taken together these results suggest that artificial G4 stabilisation in hESCs poses a barrier to CNCC differentiation. Indeed delayed differentiation was also observed when using each of two structurally distinct G4-selective stabilising molecules: N-methyl mesoporphyrin IX (NMM)[61,62] and 12459[63] (Supplementary Fig 19). This suggests that it is G4 stabilisation, rather than non-specific effects by a given molecule, that are the cause of delayed differentiation. High G4 abundance in hESCs appears to be associated with the pluripotent state, whereas dynamic changes in the G4 landscape are coupled to the transcriptional reprogramming that takes place during differentiation.

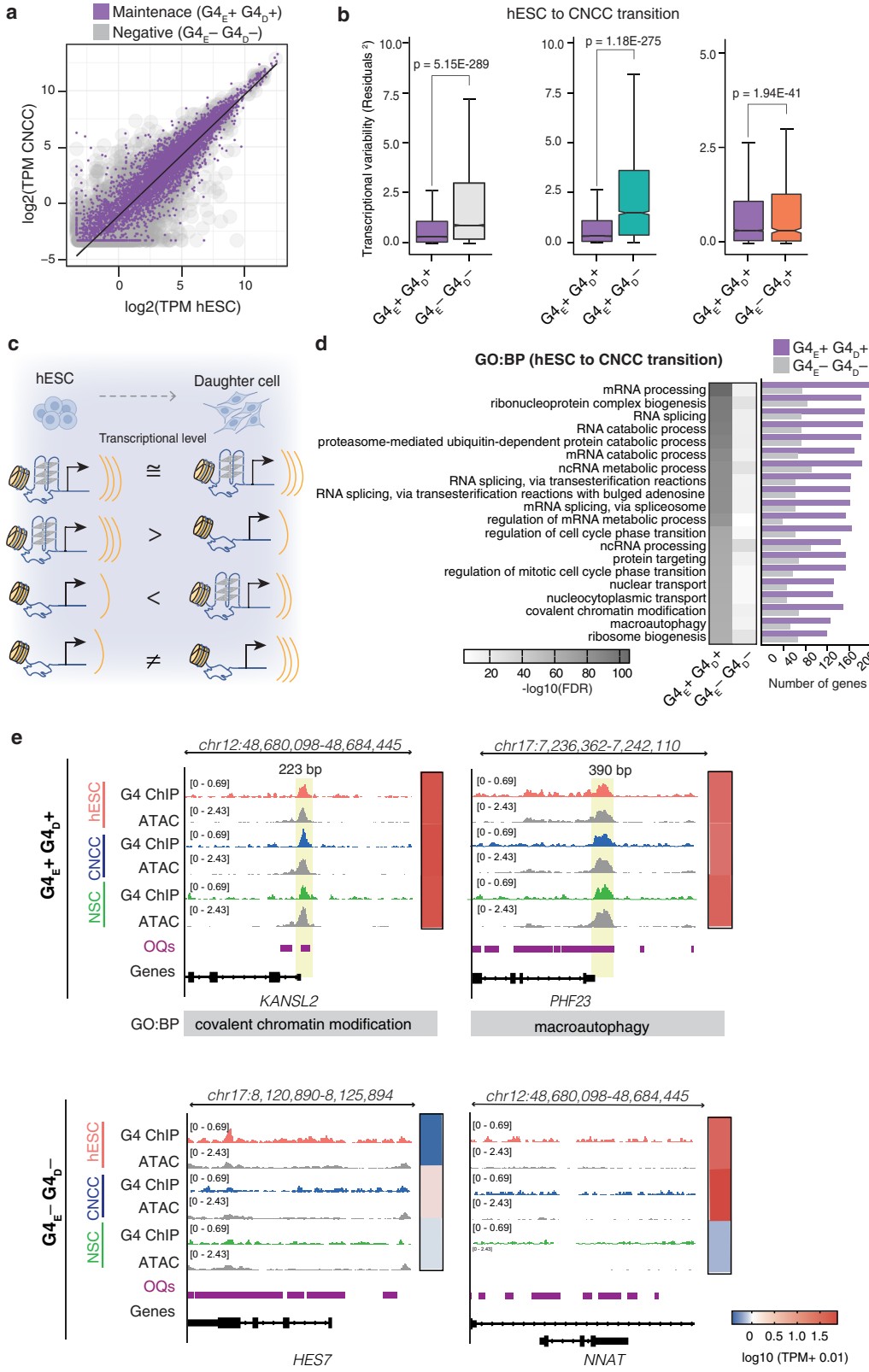

## Discussion

G4 DNA secondary structures are increasingly recognised as features with important biological functions in the mammalian genome[9,24]. Our work defines the G4 structural landscape in human embryonic stem cells. G4-ChIP-seq shows that G4s are a prominent and frequent feature in hESC chromatin. G4s are dynamically regulated upon lineage differentiation and their abundance is tightly linked to the degree of stem cell plasticity. The majority of daughter cell G4s are located at genomic loci that were also found in the embryonic state, suggestive of key common functions. During differentiation, we reveal that promoter G4 'maintenance' stabilises transcriptional levels of genes for essential cellular/homeostasis functions and is associated with important developmental changes in the histone modification

**Fig. 4 G4 maintenance during differentiation stabilises transcription of genes essential for key cellular functions. a** Scatter plot of the log2 median transcripts per million (TPM) for hESCs vs CNCCs with fitted weighted linear regression model (black line). G4$_E$; G4 promoter status in hESCs and G4$_D$; G4 promoter status in daughter cell (CNCCs). **b** Box plots of the calculated residuals of G4$_E$+ G4$_D$+ and G4$_E$− G4$_D$−, G4$_E$+ G4$_D$− or G4$_E$− G4$_D$+. Data are presented as median (centre) and interquartile range (box; the lower and upper bounds of the box represent the 25th and 75th percentiles, respectively). Whiskers represent ±1.5x interquartile range. $N = 5$ and 3 biologically independent samples for hESCs and CNCCs, respectively. Number of genes per group: G4$_E$+ G4$_D$+ (7038) and G4$_E$− G4$_D$− (16818), G4$_E$+ G4$_D$− (3515) or G4$_E$− G4$_D$+ (913). One-tailed F-test for variances. **c** Schematic depicting a summary of gene expression dynamics associated with each G4 promoter class. **d** Heatmap showing top 20 significant (FDR < 0.05) biological processes (BP) obtained from Gene Ontology (GO) enrichment analysis performed with g:Profiler for genes which are not differentially expressed (FDR > 0.05, −1 < Log2FC < 1; see Methods section) between hESCs and CNCCs and either maintain a G4 (G4$_E$+ G4$_D$+; $n = 4675$ genes) or never had a G4 (G4$_E$− G4$_D$−; $n = 3425$ genes) in their promoter. The number of genes intersecting GO term is shown in the adjacent bar plot. **e** Genome browser view for representative promoters showing G4$_E$+ G4$_D$+ or G4$_E$− G4$_D$− promoter status after hESC differentiation into CNCCs or NSCs. Yellow box highlights overlap of G4s, open chromatin sites (ATAC) and OQs[13]. Heatmap showing median gene expression (Log10(TPM + 0.01)) is also shown for the indicated genes.

landscape. In addition to highlighting G4s as a feature associated with active promoters, our work has also revealed folded G4 structures are features of many bivalent promoters. To investigate a potential causative link between G4s and lineage commitment, we perturbed the endogenous G4 landscape in hESCs using G4 stabilising small-molecule ligands. This resulted in a differentiation delay due to failure of pluripotency exit and suggests that the high abundance of G4s in hESCs relative to CNCCs and NSCs acts to maintain the pluripotent state.

Our work provides insights into how the transcription of key lineage specification and essential cellular functions/homeostasis pathways are maintained during differentiation in the face of transcriptional noise[64]. Genes that maintain a G4 promoter upon differentiation fluctuate less in expression from the hESC state, as compared to genes that lose or gain a G4. Although a similar phenomenon is seen when promoters maintain their H3K4me3 and open chromatin status alone, this effect is much weaker than that of G4s. We propose, therefore, that promoter G4s that are 'maintained' from hESCs to daughter cells upon differentiation sustain expression of associated genes with less transcriptional variability. This effect is independent of the magnitude of gene expression; thus, a consistent explanation is that the G4 may have a structural role in helping keep the underlying chromatin state permissible for transcription. This is further supported by our recent findings in transformed cells showing that promoter G4 formation precedes transcription[65]. Indeed, we found that G4s preserved from hESCs to daughter cells travel with the H3K4me3 histone modification during differentiation whereas G4 loss is correlated with loss of chromatin accessibility. Additionally, we note that stem cell G4s form not only in gene promoters but also at enhancers, super enhancers and sites of chromatin looping interactions. Thus, G4s may be potentially important elements that modulate 3D chromatin organisation to promote transcriptional consistency during differentiation.

Histone modifications are dynamic features important in the control of gene expression and differentiation[66] and here we provide evidence that G4 structures may act as an additional layer within an epigenetic regulatory system. For example, we find that hESC bivalent promoters which keep their G4 status or gain a G4 upon differentiation are more likely to transition to an active H3K4me3 promoter state in the daughter cell. Compared to bivalent promoters that lack a G4, bivalent promoters that are marked with a G4 have higher levels of H3K4me3 and thus chromatin accessibility. Our findings may extend mathematical modelling that predicts chromatin bivalency at CpGI exists as a "bistable" system, frequently switching between active and silent chromatin states[67]. For instance, G4 folding/unfolding could theoretically provide a rapid and efficient mechanism to modulate between active and silent bivalent states. Indeed, G4s have been identified as binding sites for effector proteins with histone-

modifying activities (e.g. ATRX and LSD1[9]). Recent findings in zebrafish using antisense oligonucleotides directed against predicted promoter G4s lends further support for a developmental role of G4s as the resulting embryo phenotypes were similar to those of embryos deficient in the targeted genes[68]. Further work will be necessary to understand the detailed mechanistic interplay between the G4 and histone landscapes and subsequent transcriptomic changes.

In the future, it will be important to understand how G4 formation and loss is regulated during differentiation. SWItch/ Sucrose Non Fermentable (SWI/SNF) chromatin remodeler complexes, which deconstruct the pluripotency chromatin landscape during neurogenesis[69,70] may be directly involved. Members of this family (e.g. SMARCB1 and SMARCA2) were uncovered in genetics screens for G4-interactors[71] and recent work has demonstrated that these proteins preferentially recognise G4 structures in vitro[72] and localise to G4 structures in cancer chromatin[73]. It is interesting to note that perturbations in SWI/SNF complexes contribute to 20% of human cancers[74] and is immediately suggestive of a mechanism of how G4s may become reactivated in cancer.

In conclusion, our work has highlighted important and unanticipated roles for G4 DNA secondary structures in stem cell pluripotency and cell fate specification (Fig. 6f). As G4s presage the transitioning of epigenetic and transcriptional landscapes that occur during differentiation, G4s should now perhaps be considered epigenetic features in their own right.

## Methods

**Human stem cell culture.** H9 (WA09, WiCell) and H1 OCT4-EGFP (WiCell) human embryonic stem cells (hESC) were cultured in mTESR1 media (cat. # 85850, StemCell Technologies) on six-well plates pre-coated with Matrigel (cat. # 354277, Corning). hESCs were passaged every 4–6 days at a split ratio of 1:6 to 1:12 using ReLeSR (cat. # 05872, StemCell Technologies) and mTESR1 was supplemented with 10 µM ROCK inhibitor Y-27632 (cat. # 1254, Tocris) on the first day of passage. Media was replenished daily. H9-derived neural stem cells (NSC-H9, WiCell) were cultured in StemPro NSC SFM complete medium (cat. # A1050901, ThermoFisher) on six-well plates pre-coated with Geltrex (cat. # A1569601, ThermoFisher). NSCs were passaged with StemPro Accutase using a ratio of 1:3 to 1:6. Cranial neural crest cells (CNCCs) were generated from H9-hESCs as previously described[26]. CNCCs were harvested at passages 4–5 for all subsequent downstream experiments. For each biological replicate, different passages were used for hESC and NSCs whereas different differentiations were used for CNCCs. All stem cells were cultured at 37°C in 20% $O_2$ and 5% $CO_2$. hESCs work was authorised by the Steering Committee for the UK Stem Cell Bank and for Use of Stem Cells (MRC).

**Transcription factor flow cytometry.** Human stem cells were fixed in 2% (w/v) paraformaldehyde in PBS at RT for 10 min, washed with PBS and permeabilised with PBS/0.7% (v/v) Tween 20 at RT for 15 min. Staining was performed according published procedures[75] using the following primary antibodies: rabbit (r) α OCT4 (1:200, # 2750, Cell Signaling Technology), mouse (m) α SOX2 (Clone 245610, 1:200, cat. # MAB2018, R&D Systems), m α NANOG (Clone 1E6C4, 1:1600, cat. # 4893, Cell Signaling Technology), r α PAX6 (1:50, # ab5790, Abcam) and m α Ki67 (Clone 8D5, 1:400, # 9449, Cell Signaling Technology) and secondary antibodies:

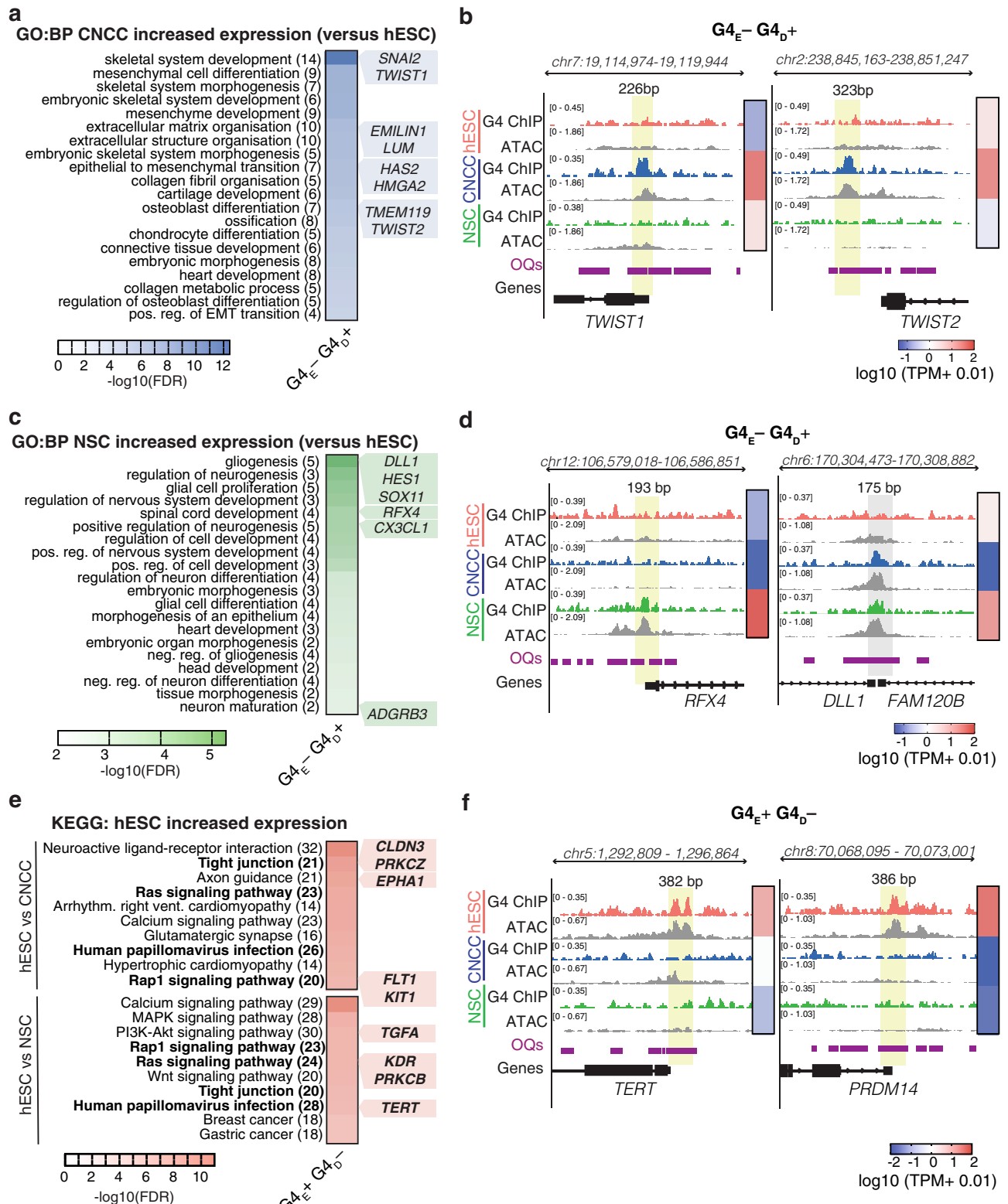

goat (g) anti-mouse 488 (1:500, cat. # A32723, Invitrogen) and g anti-rabbit 647 (1:500, cat. # A32733, Invitrogen) diluted in the blocking buffer: 1% (w/v) BSA (cat. # BP1605-100, Fisher Scientific), 10% normal goat serum (cat. # ab7481, Abcam), 0.5% Tween20 in PBS. Flow cytometry was performed on MACSquant Analyzer (Miltenyi Biotec) and data were analysed using FlowJo 10.5.3 (FlowJo LLC).

**Transcription Factor immunofluorescence microscopy.** Human stem cells were plated onto circular #1.5 coverslips pre-treated with the corresponding cell-specific

coating Matrigel (hESC), Geltrex (NSC) or 7.5 µg/ml fibronectin (CNCC) in 12-well plates. After fixing with 4% (w/v) paraformaldehyde pH 7.4 in PBS at RT for 10 min, coverslips were incubated in block solution (1% (w/v) BSA (cat. # BP1605-100, Fisher Scientific), 0.1% (v/v) Triton-X in PBS) for 1 h at 37 °C. Coverslips were then incubated with the primary antibody in blocking solution for 1 h at 37 °C, washed three times with PBS for 5 min with shaking and incubated with secondary antibody in blocking solution for 30 min at 37 °C in the dark. Coverslips were washed as before, with DAPI nuclear counterstain DAPI (0.5 µg/mL) included in the last wash and mounted onto Superfrost Plus Slides (cat. # P36961, Invitrogen)

**Fig. 5 Promoter G4 landscape changes are related to stem cell identity. a–d** Top 20 significant (FDR < 0.05) biological processes obtained from Gene Ontology enrichment analysis (GO:BP) performed with g:Profiler[84] for genes showing increased expression in **a** CNCCs vs hESCs (FDR < 0.05, Log2FC > 1) that gain a promoter G4 (G4E– G4D+; *n* = 136 genes) and **c** in NSCs vs hESCs (FDR < 0.05, Log2FC > 1) that gain a promoter G4 (G4D– G4D+; *n* = 37 genes). Number of genes in intersection for each term shown in brackets. See Supplementary Data 2 and 3 for full list of GO and KEGG terms. Genome browser view for representative gene promoters demonstrating G4E– G4D+ promoter status after hESC differentiation into **b** CNCCs or **c** NSCs. Yellow shading highlights location of cell-specific G4 across open chromatin sites (ATAC) and OQs[13]. Grey shading highlights G4s common to both CNCCs and NSCs. Heatmap showing median gene expression (Log10(TPM + 0.01)) is shown for the indicated genes. *N* = 5, 3 and 4 biologically independent samples for hESCs, CNCCs and NSCs, respectively. **e, f** Top 10 KEGG terms for genes with increased expression (FDR < 0.05, Log2FC > 1) in hESCs compared to CNCCs (*n* = 1,128 genes) or NSCs (*n* = 1175 genes) which lose a promoter G4 (G4E+ G4D−) (**e**). Genes in bold represent those in common in NSC and CNCC differentiation. **f** Genome browser view for representative gene promoters demonstrating G4E+ G4D− promoter status after hESC differentiation.

with ProLong Diamond Antifade Mountant. Images were acquired on a Nikon Eclipse TE2000E inverted microscope. Primary antibodies used: m α SOX2 (Clone 9-9-3, 1:200, cat. # ab79351, Abcam), m α NANOG (Clone 1E6C4, 1:1000, cat. # 4893, Cell Signaling Technology), r α OCT4 (1:200, # 2750, Cell Signaling Technology), m α SSEA-4 (Clone MC-813-70, 1:100, cat. #60062, StemCell Technologies), goat (g) α SOX1 (1:150, cat. # AF3369, R&D Systems), m α NESTIN (Clone 10C2, 1:00, cat. # MA1-110, Invitrogen), r α PAX6 (1:50, cat. # ab5790, Abcam), r α PAX7 (1:200, cat. # PA1-117, ThermoFisher), r α NR2F1 (1:200, cat. # PA5-21611, ThermoFisher), m α AP-2alpha (Clone 3B5, 1:100, cat. # sc-12726, Santa Cruz), m α p75 NGF Receptor (Clone NGFR5, 1:250, cat. # ab3125, Abcam) and m α phosphor-Histone H2A.X (Ser139) (Clone JBW301, 1:200, cat. # 05-636, Merck). Secondary antibodies (1:500) used: g anti-mouse 488 (cat. # A32723, Invitrogen), g anti-mouse 647 (cat. # A21236, Invitrogen), g anti-rabbit 647 (cat. # A32733, Invitrogen), g anti-rabbit 488 (cat. # A11034, Invitrogen), donkey (d) anti-mouse 488 (cat. # A32766, Invitrogen) and d anti-rabbit 594 (cat. # A21207, Invitrogen and d anti-goat 647 (cat. # A32849, Invitrogen). For staining cell surface markers (SSEA-4 and p75 NGF Receptor), coverslips were incubated with ice-cold methanol for 5 min after paraformaldehyde fixation and Triton-X omitted from the blocking buffer.

**Cell cycle analysis**. Human stem cells were fixed in 2% (w/v) paraformaldehyde in PBS at RT for 10 min, washed in PBS and permeabilised in 0.25 % (v/v) Triton-X for 5 min at RT. In all, $1 \times 10^6$ cells were then stained with DAPI (1 μg/mL in PBS) and flow cytometry used to determine cell number in G1, G2/M and S from a population of 10,000 cells (MACSquant Analyzer, Miltenyi Biotec). The percentage of cells at each stage of the cell cycle was determined using the cell cycle platform (Watson Model) in FlowJo 10.5.3 (FlowJo LLC).

**CNCC surface antigen fluorescence-activated cell sorting**. Human stem cells (hESCs, CNCCs, NSCs and mesenchymal cells) were washed with DPBS and disassociated using StemPro Accutase. After centrifugation ($300 \times g$, 10 min), cells were resuspended in Blocking buffer: DPBS/0.5% BSA (cat. # 700-104 P, Gemini Bio-Products)/2 mM EDTA buffer and stained for 10 min at 4 °C with CD266-PE (FN14, Clone ITEM-4, cat. #130104329, Miltenyi Biotec) and CD271-PE Vivo 770 (p75NTR, Clone ME20.4-1.H4) cat. #130113984, Miltenyi Biotec) previously identified in Prescott et al 2015[26] to distinguish CNCCs from hESCs, neuroectodermal spheres (NEC) and CNCC derived-mesenchymal cells. After two washes with blocking buffer, CNCCs were sorted on a FACS Aria cell sorter (BD Bioscience) using BD FACSDiva software (Version 2.0) for RNA-seq. For controls, NECs were harvested at day 4 from the CNCC-derivation protocol. CNCCs were differentiated into mesenchymal cells (smooth muscle) by culturing CNCCs in DMEM F-12 (cat. # 31330-038, Invitrogen) + 10% FBS for two weeks.

**G4-ChIP-Seq and library preparation**. G4-ChIP-seq was performed essentially as described[14,32] with the following modifications. Chromatin was isolated from 30 million human stem cells (ESC, NSC or CNCC) using the Lysis and Hypotonic Buffer for Sonication kit (cat. # 100008, Chromatrap), using 250 μL volume of each buffer, as per Step2a of the manufacturer instructions (cat. # 500239, Chromatrap). Lysed nuclei suspensions were sonicated for 25–40 cycles (30 s on/60 s on, high setting) using a water-cooled bath sonicator (Bioruptor Plus, Diagenode) until an average DNA size of 100–500 bp was reached. For ChIP, three technical replicates were performed for each biological replicate per cell line. For each ChIP, 7.5 μL of 150 ng/μL chromatin was incubated in 133.5 μL blocking buffer (25 mM HEPES pH 7.5, 10.5 mM NaCl, 110 m KCl, 1 mM MgCl₂ and 1% (w/v) bovine serum albumin (cat. #A7030, Merck)) supplemented with 3 μL 1 mg/ml RNaseA (cat. # AM2271, ThermoFisher) for 30 min at 800 rpm on a thermoshaker (Eppendorf) at 16 °C. Meanwhile, 195 μL anti-flag M2 magnetic beads (cat. # M8823, Merck) were washed three times with 1.95 mL blocking buffer and stored in 1.95 mL blocking buffer (bead blocking solution) at 16 °C on the thermoshaker (1400 rpm) for at least 1 h until required. The recombinant single-chain variable antibody BG4 was expressed via autoinduction[31]. For each ChIP, 6 μL of 4.8 μM BG4 stock was added and the reaction was incubated for 1 h at 16 °C with mixing at 1400 rpm. For each biological replicate, an input sample was processed in parallel with the omission of the BG4 antibody immunoprecipitation step. Next, 150 μL of bead solution was

added to each ChIP and incubated for 1 h at 16 °C with mixing at 1400 rpm. Beads were magnetically captured, the supernatant discarded via vacuum aspiration and beads washed four times with 400 μL 4 °C wash buffer (10 mM Tris pH 7.4, 100 mM KCl, 0.1% (v/v) Tween 20 (cat. # 11332465001, Merck)) by manual agitation in the cold room. Beads were resuspended in 400 μL wash buffer and incubated at 37 °C in a thermoshaker at 1400 rpm for 10 min. After a second warm wash, beads were resuspended in 150 μL of elution buffer (1 x TE buffer, 3 μL proteinase K (cat. # AM2546, ThermoFisher)) and incubated at 65 °C with 400 rpm shaking for 3 h. Eluted DNA was purified using MinElute Reaction Clean up Kit (cat. # 28206, Qiagen). In all, 10 ng of purified ChIP or input DNA was used for library preparation (2.5 μL Tn5 (cat. #15027865, Illumina) in tagmentation buffer (cat. #15027866, Illumina)) in total reaction volume of 40 μL with 800 rpm shaking for 20 min at 37 °C. After MinElute Reaction clean up, libraries were generated by mixing 20 μL tagmented DNA, 2.5 μL each of Nextera Index i7 and Nextera Index i5 (Nextera index primer kit, cat. # 15055290, Illumina) with 25 μLNEB Next High Fidelity 2 x PCR master mix (cat. # M0541S, NEB) in a PCR reaction using the following conditions: 72 °C for 5 min, 98 °C for 30 s followed by six cycles of 98 °C for 10 s, 63 °C for 30 s and 72 °C for 1 min. After MinElute Reaction cleanup, library quality and quantity were assessed by Bioanalyser (Agilent) and Qubit, respectively, and 12-libraries were multiplexed for next-generation sequencing on a NextSeq platform (single-end, 75 bp reads) at a final concentration of 2 pM. Each biological replicate for hESCs, CNCCs and NSCs was sonicated, ChIP performed, libraries prepared and sequenced together with additional biological replicates performed on a different day.

**ATAC-Seq and library preparation**. ATAC-seq libraries were generated from whole cells (100,000 (CNCC) or 50,000 viable cells (hESCs and NSCs)) as previously described[76]. Briefly, cells were dissociated with StemPro Accutase, washed in ice-cold DPBS and pelleted at $500 \times g$ for 5 min at 4 °C. Cells were next resuspended in transposition reaction mix (25 μL 2 x TD reaction buffer (cat. #15027866, Illumina), 2.5 μL Tn5 transposase (cat. #15027865, Illumina) and 22.5 μL sterile water)) and incubated at 37 °C in a thermoshaker at 1400 rpm for 1 h. The transposition reaction was immediately purified in 10 μL elution buffer using a MinElute Reaction clean-up kit. Tagmented DNA fragments were initially amplified using 2.5 μL each of Nextera Index i7 and Nextera Index i5 (Nextera index primer kit, cat. # 15055290, Illumina) with 25 μL NEB Next High Fidelity 2 x PCR master mix (NEB) under the following conditions: 72 °C for 5 min, 98 °C for 30 s followed by five cycles of 98 °C for 10 s, 63 °C for 30 s and 72 °C for 1 min. To reduce GC and size biases, the total PCR cycle was determined[76]. All libraries for each biological replicate were generated at the same time. The 9 libraries were multiplexed and sequenced twice on a NextSeq 500 (Illumina) with $2 \times 75$ bp paired-end reads.

**RNA-Seq and library preparation**. Total RNA was extracted from $1 \times 10^6$ cells using the RNeasy Plus Mini Kit (cat. #74136, Qiagen) as per manufacturer's instructions. RNA quality (RIN > 9) was checked via Bioanalyzer and RNA concentration determined using a Qubit fluorometer Libraries were generated from 500 ng RNA using the Truseq Stranded mRNA kit (cat. # 20020595, Illumina) as per manufacturer's instructions and sequenced on a HiSeq 4000 (Illumina) platform (50 bp single-end).

**Human reference genome and relative genomic annotations**. Human genome reference hg38 fasta file was downloaded from UCSC database (ftp://hgdownload.cse.ucsc.edu/goldenPath/hg38/bigZips/hg38.fa.gz). Genomic annotations (gtf file) were downloaded from Genecode project portal (ftp://ftp.ebi.ac.uk/pub/databases/gencode/Gencode_human/release_28/gencode.v28.annotation.gtf.gz, Release 28 GRCh38.p12).
   The reference transcriptome was obtained using Rsem (rsem-prepare-reference --bowtie2, version 1.3.1[77]). Annotations for genomic regions (i.e. exons, introns, intergenic regions, 3′UTR, 5′UTR and 58381 promoters of all coding and not coding genes defined as TSS ± 1000 bp) were extracted from the gtf file.

**Sequencing data processing, G4 and gene-expression differential analysis**. After quality controls checks with fastQC (version: 0.11.7)[78], all fastq files have

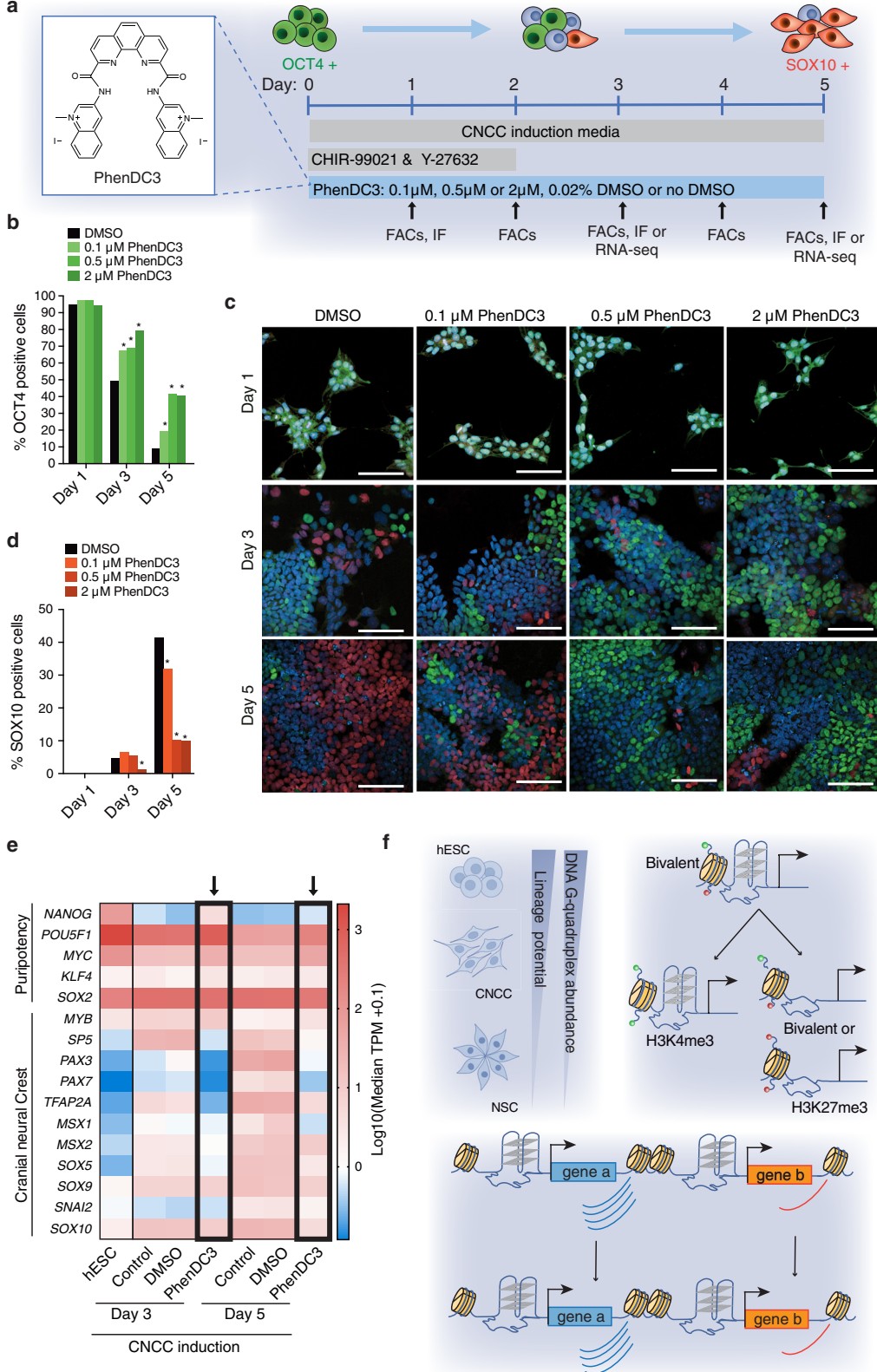

been trimmed from adaptors with cutadapt (version: 1.16[79], G4-ChIP-seq options: -m 10 -q 20 -O 3 -a CTGTCTCTTATACACATCT, ATAC-seq options: -a AGA TCGGAAGAGC -A AGATCGGAAGAG. RNA-seq options: -q 20 -m 20 -a AGA TCGGAAGAGC).

G4-ChIP-seq and ATAC-seq reads were aligned to hg38 with bwa-mem (G4-ChIP-seq: bwa mem with default options, ATAC-seq: bwa mem -M -t 12; version: 0.7.17-r1188). Next, bam files with aligned reads were generated by using samtools (samtools view -Sb -F780 -q 10 -L; version 1.8) and they were sorted. Deduplication

of the resulting mapped reads was performed with Picard MarkDuplicates (version: 2.20.3, http://broadinstitute.github.io/picard, REMOVE_DUPLICATES = true).

For ATAC-seq data, fragment size distribution was estimated using uniquely mapped reads bams with Picard CollectInsertSizeMetric. The amount of mitochondria contamination was checked by counting the number of unduplicated reads mapping to ChrM.

For both G4-ChIP-seq and ATAC-seq, regions with local enrichments (G4) or accessibility (ATAC) were identified using MACS2 (G4-ChIP-seq options: --broad,

**Fig. 6 G4 structure stabilisation delays hESC differentiation. a** Schematic of G4-ligand treatment experiment: OCT4-EGFP expressing hESCs were differentiated into CNCCs using a 2-day pulse using GSK3β (CHIR99021) and ROCK inhibition (Y-27632)[59]. Samples were taken at the indicated times to determine the percentage of cells positive for the pluripotency marker OCT4 (by flow cytometry and immunofluorescence microscopy (IF)) or SOX10 (by IF; lineage marker for CNCC) or RNA-seq. Insert chemical structure of the G4-specific ligand PhenDC3. **b–d** Proportion of **b** OCT4 and **d** SOX10 positive cells determined in IF studies. $N = 52$ fields of view per sample. One biological replicate (see Source Data for detailed cell counts). *: $p < 0.05$, one-sided Pearson's $\chi^2$ test for proportions (see Source Data for exact p-values and number of cells analysed). See Supplementary Fig. 17 for additional biological replicates. **c** Representative confocal IF images of differentiating OCT4-EGFP hESCs treated with either DMSO or PhenDC3. SOX10 = red, OCT4 = green and DAPI (nuclear stain) = blue. Scale bar = 100 μm. **e** Heatmap showing gene expression (Log10 median gene expression (TPM + 0.1)) of the indicated pluripotency and cranial neural crest genes in hESCs and hESCs treated with cranial neural crest induction media alone (Control) or with DMSO or 2 μM PhenDC3 at 3 and 5 days of differentiation. **f** Schematic of the main findings of our study: G4 abundance correlates with stem cell plasticity (top left); G4 presence is associated with particular promoter histone landscapes in differentiation (top right); and maintenance of a promoter G4 from hESCs to differentiated daughter cells preserves the transcriptional output of a gene independent of transcription levels (bottom).

ATAC: default options; version: 1.2.2). For G4-ChIP-seq, local enrichments for each biological replicate were called from separate input control libraries. G4 consensus regions of each biological replicate were obtained as the regions observed in 2 out of the 3 technical replicates (multiIntersectBed; using bedtools version: 2.27.4). The final cell-type consensus (G4 and ATAC respectively) was obtained by selecting regions reproducibly observed in 2 of the 3 biological replicates. Genome-wide track files reporting reads per million (RPM) of G4 and ATAC signal were obtained by quantifying the genome read coverage across the binned regions (10bp-bins) and scaling it to a factor reflecting the individual library size (deeptools bamCoverage -bs 10 –scaleFactor 1,000,000/Lib_size[80], version 3.3.0). The similarity of G4 signal across individual libraries from all three cell types were evaluated with hierarchical clustering of RPM G4 estimated at regions obtained by unifying all three consensus regions (Euclidean distance, agglomerative strategy: ward.D3). Whenever we refer to G4 or ATAC signal in a specific cell type, individual libraries of the same cell type were combined together by calculating the median RPM signal.

Cell G4 consensus regions were compared to the same cell ATAC consensus (ATAC) and to in-vitro observed G4 quadruplex (OQs[13]) by evaluating the percentage of overlaps.

Differential G4-binding was done with edgeR (version 3.26.8). Library size and read coverage at a multi-cell G4 consensus (merging all three cells individual G4 consensus) regions were computed. Then, a generalised linear model (glmLRT) with default parameters (negative binomial log-linear distribution of read counts) was used to assess regions with the differential binding signals. Specifically, each cell was compared to one of the other two in turn and regions with differential signal were those with FDR ≤ 0.05.

RNA-seq trimmed reads were aligned to hg38 by using rsem (rsem-calculate-expression --phred33-quals -p 40 --bowtie2, version 1.3.1[77]) and files containing genes and isoform level expression estimates were produced. Genes level expression estimates files (genes.results) were used for the subsequent analysis. When technical replicates were present, TPM (transcript per million) from the expected counts from the gene levels estimate files (genes.results) were averaged. As a quality control, the similarity across technical replicates was assessed by computing correlation between TPM and Pearson correlation was $R > 0.95$. Similarity across biological replicates was assessed by hierarchical clustering (experiments) paired to k-mean (promoters, with $k = 2$) by using Heatmap from the complexHeatmap R package (hierarchical clustering using Euclidean distance and average linkage, R version: 3.6.1 (2019-07-05)).

The differential gene expression analysis between each pair of cells (hESCs versus CNCCs, hESCs versus NSCs and NSCs versus CNCCs) was performed with EdgeR[81] using the expected_counts from the gene.results files generated during the alignment step. For each pairwise comparison, genes with differential expression (FDR < 0.05 and abs(log2FC)>1) were called using the function glmQLFit taking into account the presence of multiple biological replicates in the experimental design matrix (5 for hESCs, 3 for CNCCs and 4 for NSCs). For each cell type expression levels have been summarised as the median TPM observed across all the biological replicates. The differential gene expression analysis for samples in the PhenDC3 experiment was performed as above with differential expression cutoff FDR < 0.05 and abs(log2FC)>2.

**External histone modifications ChIP-seq processing.** Histone modification ChIP-seq maps were downloaded from the Sequence Read Archive database (SRA) using fastq-dump (version: 2.10). Adaptor trimming and alignment was performed similarly to the G4-ChIP-seq. The list of data used by cell type is the following:

hESCs:

- H3K4me3: GSM602296
- H3K27me3: GSM602293
- Input: GSM602292
- H3K4me1: GSM602295
- H3K9me3: ENCFF112ULZ [https://www.encodeproject.org/experiments/ENCSR972SMV/]

CNCCs:

- H3K27me3: GSM1817179
- H3K4me3: GSM1817174
- H3K4me3: GSM1817175
- H3K4me3: GSM1817176
- Input: GSM1817222
- H3K4me1: GSM1817170

NSCs:

- H3K27me3: GSM818033
- H3K27me3: GSM818032
- H3K4me3: GSM767350
- H3K4me3: GSM767351
- Input: GSM767355
- Input: GSM767356
- H3K4me1: GSM602303
- H3K9me3: ENCFF452NFM [https://www.encodeproject.org/experiments/ENCSR800IIW/]

Peak calling was performed with macs2 (default options for H3K4me3, --broad option for H3K27me3). When replicates were present, the consensus regions were defined as loci observed in at least two replicates. Genome-wide track files were generated by normalising read coverage by library size (RPM). Summarised RPM signal per cell type and mark was calculated as the median RPM G4 signal across replicates.

**Other external data used for comparative analysis.** Maps used for comparative analysis were taken from previous studies and public databases. When needed, genomic locations were converted into bed files and genomic coordinates from previous genome versions were lifted to hg38 by using USCS liftOver tool (command line at: https://genome-store.ucsc.edu).

- Enhancer maps- H3K27ac regions[6,26,81] defined as distal to promoter regions (not-overlapping with promoter):

  - hESCs[81]: GSM602294
  - CNCCs[26]: GSM1817151, GSM1817152, GSM1817153
  - NSCs: Dataset shared with permission from Akshay Bhinge from the publication[82] (https://pubmed.ncbi.nlm.nih.gov/24802670/)

- Master transcription factor binding sites:

  - hESCs:
    OCT4: GSM2816629
    NANOG: GSM2816625
    KLF4: GSM2816627

  - CNCCs:
    NR2F1: GSM1817190
    TFAP2A: GSM1817197

  - NSCs:
PAX6: Dataset shared with permission from Akshay Bhinge from the publication[82] (https://pubmed.ncbi.nlm.nih.gov/24802670/)

- Histone acetyltransferase p300

  - hESC: GSM602291
  - CNCC: GSM1817181

- Methylated or not CpG islands (hESCs)

- Processed Table used (Dataset S5 in publication[83])

- Chromatin architectural proteins (hESC)[84]:

  CTCF: GSM2816619
  NIPBL: GSM2816642
  RAD21: GSM2816615
  RAD21:GSM2816616
  RINGB1: GSM2816631
  ZNF143; GSM2816621

- 3D contact regions from promoter-capture Hi–C data:

  - hESC[40]: Processed data used from Supplementary Table 1 (http://osf.io/sdbg4). Raw data: GSE86821
  - NSCs[41]: GSE86189

**Characterisation of G4 fold-enrichments at sites of interests**. G4 fold-enrichments over random chance were evaluated at various sites of interest by using the Genomic Association Tester (GAT, https://gat.readthedocs.io/en/latest/contents.html, 1000 randomisations) restricting the analysis the human whitelist.

**Promoter annotations**. Promoters were annotated based on various criteria: epigenetic data, G4 enrichment data in individual cell types and transitions of G4 enrichments from hESCs to one daughter cell (CNCCs or NSCs).

Promoter annotations based on epigenetic maps:

- H3K4me3: promoter overlaps with at least one H3K4me3 region;
- H3K27me3: promoter overlaps with at least one H3K27me3 region;
- Bivalent: promoter overlaps with regions where both H3K4me3 and H3K27me3 colocalise;
- Other: promoter overlaps with both H3K4me3 and H3K27me3 but the two marks do not colocalise;
- Unmarked: promoter does not overlap with H3K4me3 or H3K27me3.

Promoter annotations based on G4 enrichments (in cell of interest):

- G4 positive: promoter overlaps with at least one G4 cell-consensus peak.
- G4 negative: promoter does not overlap with any G4 cell-consensus peaks.

Promoters annotations describing how G4 enrichments transitioned from hESC (E) to each of the two daughter cells (D: daughter i.e., CNCC or NSC):

- $G4_E + G4_D-$: G4 in hESC, no G4 region in daughter cell (G4 is lost in daughter cell).
- $G4_E + G4_D +$: G4 in both hESC and daughter cell (G4 is maintained from hESC to daughter cell).
- $G4_E - G4_D +$: no G4 in hESC, G4 region in daughter cell (G4 is acquired in daughter cell).
- $G4_E - G4_D-$: no G4 regions in hESC and daughter cell.

Integration of expression levels, G4 signal levels, histone modification overlaps and G4 enrichment overlaps across all three stem cell types. Promoters annotations were combined into a single matrix together with:

- median TPM across biological replicates for each individual cell type;
- median RPM G4 signal at across replicates;
- number of overlaps between promoter region and each of the three consensus G4 maps (hESCs, CNCCs, NSCs);
- number of overlaps of various epigenetic maps (H3K4me3, H3K27me3 regions) with the promoter (0: no overlap, N: number of overlaps).

All pie charts and alluvial plot are derived by querying this matrix with different filters. Difference in proportions was tested by using the proportion test based on the chi2 statistic (R function prop.test(), Pearson's $\chi^2$ test for proportions).

**Analysis of gene expression levels across promoter classes**. Differences in expression levels across promoter classes (epigenetic and G4-based classification) were assessed. Genes were first defined by their "epigenetic"-classes (H3K4me3, H3K27me3, bivalent and unmarked) and subsequently stratified by the presence or absence of promoter G4 (G4+ and G4−) resulting in a total of distinct 8 groups. For each of the eight groups, we tested difference between the distribution of expression levels (median TPM) by using the Kolmogorov–Smirnov test (ks.test() function in R).

To assess whether gene expression levels directly recapitulate the G4 landscape, gene expression (median TPM) levels and G4 signal (median across replicates) at the promoters of the entire set of 58381 genes were calculated. Specifically, for each gene, a 6 element-long vector (one expression value for each of the 3 cells and one G4 promoter signal level for each of the 3 cells) was built. Values above 75th quantile (by column) were capped to control for outliers' effects and the final matrix was standardised by column (z-score). K-medoid clustering of the genes

identified six different groups. To summarise global expression levels and G4 levels distributions in each cluster, 6 sets of boxplots were produced (3 boxes per cluster, where each box represents one stem cell type).

**Analysis of transcriptional stabilisation**. Transcriptional stabilisation and variability of individual genes between hESC (E) and differentiated cells (D, daughter) were explored in relation to the G4 promoter signatures with two orthogonal approaches.

Firstly, gene expression (TPM) levels were directly compared between hESCs and daughter cells. Only genes with TPM > 0 in at least one of the two cells under investigation were considered. For each promoter group ($G4_E+G4_D-$, $G4_E+G4_D+$, $G4_E+G4_D+$, $G4_E−G4_D−$), we fitted a weighted linear regression to model the relationship between the two sets of expression levels. The weights used in the fitting are expression levels of hESCs that represent conceptually the reference starting condition. Residuals of the data from the regressed model were computed and used to quantify the spread of the transcriptional variability. After each fitting step, the coefficient of "goodness of fit" $R^2$ was computed and the F-test was used to assess if there were significant differences in transcriptional stability between pairs of promoter groups. A similar analysis was performed using our ATAC-seq data and published H3K4me3 ChIP-seq data[6,83]. In each of the two cases, promoters were divided into four groups based on the presence/absence of the mark at promoters in hESCs/daughter cells. The ranking of $R^2$ values was used to cross compare and determine which feature (G4, accessibility or H3K4me3) had a greater impact on stabilising expression when the feature is transmitted from hESCs to daughter cells.

Secondly, transcriptional stabilisation was analysed using the outcome of the differential gene expression. For the 4 promoter classes, the proportion of genes belonging to each of the following groups was determined:

1. genes differentially up-regulated in hESC (FDR < 0.05, log2FC[daughter/hESC]< -1);
2. genes differentially up-regulated in the daughter cell (FDR < 0.05, log2FC[daughter/hESC] >1),
3. genes not differentially expressed (FDR > 0.05, −1 < log2FC[daughter/hESC] <1).

We obtained three proportions and tested differences (R function prop.test(), Pearson's $\chi^2$ test for proportions).

**Gene ontology and gene enrichment analysis**. G:Profiler[84] web interface was used to determine enriched Gene Ontology (GO), Biological Processes (BP) terms, KEGG pathways, Reactome pathways and Wikipathways using the g:GOSt Functional Profiling analysis (using right-sided (enrichment) hyper-geometric test and Benjamini–Hochberg adjustment of the resulting p-value). All known genes and only experimental evidence codes (EXP, IDA, IPI, IMP, IGI, IEP) were used. Term size was limited to 20–450 genes.

**G4 de novo motif discovery**. De novo motif discovery was performed on the FASTA sequences (bedtools getfasta) extracted from the G4 consensus regions (pre-filtered 5–2000 bp) using MEME-ChIP[85] web-interface (options: Enrichment mode Classic; Set of known motifs: Eukaryote DNA, Human and Mouse (HOCOMOCO v11 FULL), order-1Background, MEME Site Distribution: 0 or 1 occurrence, MEME motif count: 3 and MEME Motif width: 6–30 wide (inclusive). Occurrences of selected motifs across G4 consensus regions were quantified by performing a dedicated search with FIMO, Find individual motif occurrences[86]. Density was calculated by dividing motif occurrence/residues.

**G4 immunofluorescence microscopy**. Human stem cells were plated onto removable three-well chamber slides (cat. # 80381, Ibidi) pre-treated with the corresponding coating: Matrigel (hESCs), Geltrex (NSCs) or 7.5 µg/ml fibronectin (CNCCs). Cells were fixed in 4 % (w/v) paraformaldehyde pH 7.4 in PBS and permeabilised with PBS/0.1 % (v/v) Triton-X, with incubation for 10 min at RT for each step. Slides were blocked with 5% normal goat serum (cat. # ab7481, Abcam), 0.5% Tween20 in PBS for 1 h at 37 °C and then directly incubated with 50 nM BG4 in blocking buffer for 1 h at 37 °C. After 3 × 5 min washes with PBST (0.1 % (v/v) Tween20), slides were incubated with rabbit anti-FLAG (1:800, cat. # 2368, Cell Signaling Technology) for 1 h at 37 °C. After washing, slides were incubated for 30 min at 37 °C with goat anti-rabbit 647 secondary antibody (1:500, cat. # A32733, Invitrogen). Following three 5 min washes with PBST (DAPI counterstain (0.5 µg/mL) was included in the second wash), chambers were removed and slides mounted to #1.5 coverslips using ProLong Diamond Antifade Mountant (cat. # P36961, ThermoFisher). 3 biological replicates per cell line were performed. Confocal z-stacks images (13 steps) were acquired using a Leica TCS SP8 microscope with a HC PL APO CS2 1.4 NA ×100 oil objective (Leica Microsystems) at a scan speed of 400 Hz and sampling rate of 0.11 µm × 0.11 µm × 0.30 µm. The DAPI channel was excited using 405 nm diode laser (at 405 nm) and the white-light pulsed laser (SuperK EXTRENE, NFT Photonics) was used to excite the secondary antibody fluorophore (at 647 nm). Fluorescence detection as performed sequentially with hybrid detectors (Leica HyD Photon Counter) at wavelength ranges of 657–733 nm and 415–525 nm for Alexa Fluor 647 and DAPI, respectively. The pinhole was set to one airy unit and laser power and gain settings were kept

constant between samples and biological replicates. Images were deconvolved using Huygens Professional Software (Scientific Volume Imaging BV). Representative images were processed in the open source imaging software FIJI[87] (version 2.0.0-rc-69/1.52p) and ICY (version 1.0; http://icy.bioimageanalysis.org)[88] and assembled using Adobe Illustrator 2019. G4 signal density (sum G4 signal/ sum DAPI signal per nucleus), calculated on the sum projection of the Z-stack, was performed using FIJI followed by ICY. First in FIJI, nuclear regions of interest (ROIs) were generated from the DAPI channel using a Gaussian Blur (sigma = 3), Huang dark automatic thresholding and Watershed function to separate touching objects. Nuclei that did not segment properly were removed manually using the analyse particle function. ROIs were then used in the ICY protocol (ICY_microscope_image_analysis_file found on lab github website) using all specified parameters to extract G4 and DAPI signals.

**G4 selective ligands**. The bisquinolinium phenanthroline derivative PhenDC3 was synthesised in-house according to the protocol outlined in De Cian et al.[60]. The triazine derivative 12459 was synthesised in-house according to the protocol outlined in Douarre et al.[63] and N-methyl mesoporphyrin IX (NMM)[61,62] was purchased from Merck cat. #258806. All G4 ligands were dissolved in DMSO to 10 mM stock solutions.

**IncuCyte live-cell analysis after G4 ligand treatment**. H1-hESC OCT4-EGFP were grown in 12-well plates for 5 days (144 h) in the presence or absence of PhenDC3 (313 nM, 625 nM, 1.25 µM, 2.5 µM and 5 µM) or equal volume of DMSO. DMSO concentration (0.02%) was kept consistent between PhenDC3 and DMSO-treated wells. Cell growth was monitored using IncuCyte ZOOM live cell analysis (Sartorius) and cell confluency was calculated as a percentage of the well area covered. Scans were performed every 3 h: nine scans per well. Media was changed daily.

**5-day CNCC generation**. CNCCs were also generated from H1-hESC OCT4-EGFP (or H1-hESC (WiCell) for Supplementary Fig. 19) using a 5-day procedure as previously described[59]. H1-hESC OCT4-EGFP were grown at 37 °C in 5% $O_2$ and 5% $CO_2$ for at least three passages prior to the start of differentiation. At the step of cell seeding in the procedure the following method was utilised to ensure that cellular incubation with ChIRON 99021 and the G4-selective ligand (PhenDC3, NMM or 12459) or DMSO alone occurred simultaneously: $4 \times 10^4$ cells/well (in 0.5 mL neural crest induction media: 1 x DMEM-F12 (cat. # 31330-038, Invitrogen), 1x B27 supplement (cat. # 17504044, ThermoFisher), 1X Glutamax supplement (cat. # A1286001, ThermoFisher), 0.5% (w/v) BSA (cat. # 700-104 P, Gemini Bio-Products), supplemented with 10 µM ROCK inhibitor Y-27632) were seeded onto Matrigel-coated 24-well plates (for flow cytometry) and removable three-well chamber slides (cat. #80381, ibidi) or µ-Plate 24-well (cat. #82406, ibidi) (for immunofluorescence) already containing neural crest induction media (0.5 mL/well for three-well chamber slides or 0.25 mL/well for 24-well plates) supplemented with 10 µM ROCK inhibitor Y-27632, 2 x concentration of ChIRON 99021 (6 µM) (cat. #CT99021, Selleck) and 2x concentration of G4 ligand (i.e. PhenDC3: 4 µM, 2 µM, 1 µM, 0.2 µM; NMM: 4 µM, 2 µM, 1 µM and 12349: 4 µM, 1 µM, 0.2 µM) or DMSO only. For RNA-seq, 60 mm culture dishes were used with $4.2 \times 10^5$ cells/plate seeded in a total of 4 mL media. ROCK inhibitor Y-27632 and ChIRON 99021 was included in the induction media for the first 48 h. Induction media was changed daily until the day of collection for further analysis. Independent differentiations were set up for flow cytometry and IF experiments.

**OCT4-EGFP flow cytometry**. OCT4-EGFP expression was monitored using MACSquant Analyzer (Miltenyi Biotec) and MACSquantify Software (ver: 2.11.5) for each day of the 5-day neural crest differentiation protocol. Live cells were washed twice in DPBS, dissociated with StemPro Accutase and resuspended in neural induction media. The sample was subjected to flow cytometry using the blue laser (488 nm) and B1 (525/50 nm) channel and data was analysed via FlowJo 10.5.3 software (FlowJo LLC). Overton % positive statistic was calculated using the 'compare populations' function and the significance of the OCT4-EGFP shift was determined using the $\chi^2$ test. Population shifts between PhenDC3- and DMSO-treated cells were considered significant ($p < 0.01$) if the change ($\chi^2$ value) was greater than what was observed between DMSO (0.02% DMSO) and untreated control (no PhenDC3 or additional DMSO).

**Immunofluorescence microscopy for 5-day neural crest differentiation experiment**. Immunofluorescence of H1-hESC OCT4-EGFP cells undergoing cranial neural crest differentiation was performed according to the protocol outlined in section "Immunofluorescence Microscopy", however, primary antibody incubation was performed overnight at 4 °C. Primary antibodies: m α OCT4 (Clone GT486, 1:400; cat. # ab184665, Abcam), G α SOX10 (1:200; cat. # AF2864, R&D Systems) R α phospho-53BP1 (Ser1778) (1:300; cat. # 2675, Cell Signaling Technology). Secondary antibodies: d anti-mouse AlexaFluor 488, d anti-goat AlexaFluor 647, d anti-AlexaFluor 594 (ThermoFisher). Slides were imaged on the Operetta CLS High-Content Analysis System (PerkinElmer) and analysed using Harmony High-Content Imaging and Analysis Software (PerkinElmer). For each well of an ibidi slide (for results in Fig. 6), 12 planes (0.5 µm apart) in 52 fields were acquired using 40x Water Objective (two peak Autofocus and binning 2) with the following exposure times: DAPI (20 ms, power-

80%), AlexaFluor488 (500 ms, power 100%) and AlexaFluor647 (240 ms, power-100%). Analysis sequence using Harmony High-Content Imaging and Analysis software (version 4.9; PerkinElmer) was performed using the following analysis blocks: Input Image block: Flatfield Correction – Basic; Brightfield Correction- yes, Stack processing- 3D analysis. Filter Image block: Channel -DAPI, Method-Sliding Parabola, Curvature – 5. Find Nuclei block: Channel – DAPI, Method – C (Common Threshold 0.11, Volume >120 µm³), Output – Nucleus. Calculate Position Properties block: Image Border Distance, Output – Nucleus. Select Population block: Nucleus Distance to border – 4, Output- Nuclei removed edges. Calculate Intensity Properties block: Input-Nuclei removed edges, Channel-AlexaFluor647, Method- standard, Property Intensity Nucleus AlexaFluor647. Calculate Intensity Properties block: Input- Nuclei removed edges, Channel- AlexaFluor488, Method- standard, Property Intensity Nucleus AlexaFluor488. Calculate Morphology Properties block: Volume, Object Height. Define Results: Intensity Nucleus Sum Alexa 488, Intensity Nucleus Sum Alexa 647, Volume, Object Height. OCT4 and SOX10 intensity were calculated per nucleus (AlexaFluor488 sum/Volume and AlexaFlour647/Volume). For results in Supplementary Fig. 19, 5 planes (1.2 µm apart) in 40 fields were acquired using 20x Air Objective NA 0.8 (one peak Autofocus and binning 2) with the following exposure times: DAPI (40 ms, power 20%), AlexaFluor488 (600 ms, power 100%), AlexaFluor594 (400 ms, power 100%) and AlexaFluor647 (100 ms, power- 100%) for each well of the µ-Plate 24-well. Analysis sequence was performed as above, taking into consideration the additional AlexaFlour594 channel. The percentage of OCT4 or SOX10 or 53BP1 positive cells was calculated by determining the number of cells that have median fluorescence intensity above the background threshold as defined by the secondary antibody only control (threshold was determined for each plate individually).

**Western blotting**. Cells were grown in 6-well plates and harvested on ice using 100 µl RIPA buffer (SERVA Electrophoresis GmBH, cat #39244) supplemented with Halt Protease Inhibitor Cocktail (100X) (ThermoFisher Scientific, cat #79438) by cell scraping. The cell lysate was next sonicated for 5 cycles (30 s on/ 60 s on, high setting) using a water-cooled bath sonicator (Bioruptor Plus, Diagenode) and centrifuged at $20,000 \times g$ at 4 °C for 30 mins. Subsequent supernatant was collected, and protein concentration was measured by Direct Detect Spectrometer (Merck). Capillary electrophoresis in a Wes Protein Simple Western System (ProteinSimple) was then performed according to (using a final protein concentration of 8 mg/mL) to the manufacturer's protocol (https://proteinsimple.com/) using an anti-rabbit detection module (25-capillary 12-230 kDa; ProteinSimple, cat #SM-W004 and detection kit (ProteinSimple, cat #DM-001) and corresponding anti-rabbit primary antibodies: CHK1, 1:500 (Proteintech, cat #A5887-1-AP), phospho-CHK1 (Ser345), 1:50 (Cell Signaling Technology, cat #2341); CHK2 1:250 (Proteintech, cat #a3954-1-AP); phospho-CHK2 (Thr68) 1:250 (Cell Signaling Technology, cat #2197) and GAPDH 1:50 (Clone D16H11, Cell Signaling Technology, cat #5174 S). Protein bands were quantified as the area-under-the-curve using the Compass software (ProteinSimple, Version 5.0.1). Three biological replicates were performed, see Source Data for calculations.

**RT-qPCR**. In all, 1 µg of RNA was reverse transcribed using SuperScript IV VILO Master Mix Enzyme (11766050, Invitrogen), following the manufacturer's instructions (without gDNA digestion). qPCR reactions were set up with 5 µl Fast 2X SYBR Green Master Mix (Thermo Fisher Scientific), 2.5 µl cDNA diluted 1:50 in nuclease-free water (AM9937, Ambion), and 2.5 µl 1 µM primer mix, and performed using a C1000 Touch thermal cycler (CFX96 and CFX384, Bio-Rad) with the following programme: 95 °C for 20 s, 95 °C for 3 s, 60 °C for 30 s and 70 °C for 1 s, with steps 2–4 repeated for a total of 40 cycles. Melting curves were measured using 0.5 °C step from 65 °C to 95 °C. Three technical replicates were performed for each reaction, and expression was normalised to GAPDH using the ΔΔCt method in Bio-Rad CFX Manager software (version 3.1). Primer sequences can be found in Supplementary Data 4.

**G4-ChIP-qPCR**. G4-ChIPs were performed as described in the "G4-ChIP-Seq and library preparation" method section. To perform ChIP-qPCRs, ChIP samples then were diluted 1:10 and input samples were diluted 1:20 in nuclease-free water (AM9937, Ambion). qPCRs were performed using the PCR conditions described in "RT-qPCR". Percentage recovery was calculated as (100×2^(adjusted input Ct-average ChIP Ct)), and enrichment was calculated vs TMCC1 (negative control for G4 formation) as previously described in the Nature Protocols paper Hansel-Hertsch et al.[32]. Primer sequences can be found in Supplementary Data 4.

**Statistics and reproducibility**. No statistical methods were used to predetermine sample sizes. The number of replicates for each experiment and statistical tests used to analyse data are reported in the appropriate figure legends and methods sections.

**Reporting summary**. Further information on research design is available in the Nature Research Reporting Summary linked to this article.

## Data availability
The data that support this study are available from the corresponding author upon reasonable request. The G4-ChIP-seq, ATAC-seq and RNA-seq data generated in this study have been deposited under the accession code GSE161531. The RNA-seq generated

from the PhenDC3 differentiation experiment has been deposited under the accession code GSE166246. Imaging datasets are available from the corresponding author on reasonable request - the full 3D confocal images are extremely large in size. Source Data are provided with this paper. Processed data has been made available at: https://github.com/sblab-bioinformatics/G4_in_stem_cells_diff. The following previously published datasets were also used: GSM602296, GSM602293, GSM602292, GSM602295, ENCFF112ULZ, GSM602294, GSM2816629, GSM2816625, GSM2816627, GSM602291, GSM2816619, GSM2816642, GSM2816615, GSM2816616, GSM2816631, GSM2816621, GSE86821, GSM1817179,GSM1817174, GSM1817175, GSM1817176, GSM1817222, GSM1817170, GSM1817151, GSM1817152, GSM1817153, GSM1817190, GSM1817197, GSM1817181, GSM818033, GSM818032, GSM767350, GSM767351, GSM767355, GSM767356, GSM602303, ENCFF452NFM [https://www.encodeproject.org/experiments/ENCSR800IIW/], GSE86189. Source data are provided with this paper.

## Code availability

Code is available on the lab github webpage https://github.com/sblab-bioinformatics/G4_in_stem_cells_diff.

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

## Acknowledgements

The Balasubramanian laboratory is supported by Cancer Research UK core (C9545/A19836) and programme award funding (C9681/A29214) and Herchel Smith Funds. S.B. is a Senior Investigator of the Wellcome Trust (209441/Z/17/Z; S.B.). S.F. is also supported by the Leverhulme Trust (ECF-2021-398). We thank the staff of the Genomic, Light Microscopy, Flow Cytometry and Research Instrumentation and Cell Services core facilities at Cancer Research UK Cambridge Institute. We specifically acknowledge the Genomic Core faculty for the generation and sequencing of the RNA-seq libraries. Thanks to Dr. Sara Prescott, Dr. Maneeshi Prasad and Dr. Martin Garcia-Castro for advice on the generation and maintenance of human cranial neural crest cells; Dr. Ilaria Falciatori for human embryonic stem cell culture support and Balasubramanian lab members for discussion.

## Author contributions

Conceptualisation by K.G.Z., D.T. and S.B.; methodology by K.G.Z., A.S. and D.T.; formal analysis by K.G.Z., A.S., G.M. and G.P.; investigation by K.G.Z., S.F. and C.D.; resources by S.A.; data curation by A.S; writing-original draft by K.G.Z., A.S. and D.T.; writing – review & editing by K.G.Z., A.S., S.F., D.T. and S.B.; visualisation by K.G.Z., C.D. and A.S.; supervision by D.T. and S.B.; project administration by D.T. and S.B.; funding acquisition by S.B.

## Competing interests

S.B. is a founder and shareholder of Cambridge Epigenetix Ltd. All other authors declare no competing interests.
