## [Peer Review File · Nature Communications]

REVIEWER COMMENTS

Reviewer #1 (Remarks to the Author):

The work entitled 'G-quadruplex DNA structures in human stem cells and differentiation', by Zyner and colleagues presents results suggesting that the G4s are important in cell differentiation. Results show that i) G4 are abundant in embryonic stem cell promoters and enhancers and are lost during cell differentiation; ii) bivalent promoters conserving or gaining G4 maintain positive methylation marks and have fewer fluctuations in their expression, correlating well with the genes that should or should not be expressed as cell differentiation proceeds; and, iii) the stabilization of G4 prolongs the stem cell features.

The work is original and of great significance to the study of the biological role of G4s, as well as to enlarge the knowledge regarding the molecular mechanisms responsible for cell differentiation. Most of the conclusions and claims are supported by the experimental pieces of evidence, which were properly analyzed and interpreted. However, additional experiments and analyses might strengthen the conclusions (see below). The methodology is correct and has been clearly provided, making it possible to reproduce.

Major concerns:

Throughout the work, the authors have used the words "development" and "differentiation" interchangeably. In this work, it is more appropriate to refer to the ESC-NSC and ESC-CNCC processes as cell differentiation processes. On the other hand, although ESC-NSC and ESC-CNCC differentiation take place during early development, it is not correct to state that experiments done in cultured cells represent 'early developmental' events.

The main problem of the work is the use cNCC and NSC as cell lines showing different lineage potential. NSCs do not derive from cNCC. The opposite could be possible; i.e., some NSC could derive in NCC (but no cranial). So, NSCs could be even less differentiated than cNCC. Figure 1 has to be changed to solve this problem. Experiments in which ESCs are induced to CNCC (as was done in some experiments in this work), and then those CNCCs are induced to any of the potential derivatives of CNCC (e.g. chondrocytes) could be more appropriate to compare cell differentiation stages.

The word "inheritance" indicates that the G4s are "conserved" as DNA structure during cell differentiation. What is the experimental evidence to sustain that the G4s do not unfold and refold during the differentiation process? In a similar fashion, the word "maintenance" was used in several phases (e.g., line 167), but experimental data supporting this suggestion are scant. In my opinion, it is not correct to use the words inheritance/transmission/propagation since these words suggest that G4 cannot unfold/refold during cell differentiation.

A direct check (validation) would be lacking for characteristic genes of each cell type analyzed, both in cases in which the G4 status is maintained (G4E- G4D- or G4E + G4D +) and in those that change (G4E- G4D + or G4E + G4D-) and the corresponding expression levels.

Does the stabilization of G4 by PhenD3, which apparently delays the stem feature, increase the levels of H3K4Me3? Some other drug should be used in similar experiments, even if it damages the DNA a little. It is important to rule out unspecific PhenD3 effects. Alternatively, the effect of G4-helicase depletion could be tested. Hypothetically, G4-helicase depletion might lead to similar results. Even more, the overexpression of G4-helicases should promote cell differentiation.

The expression of some of the genes having bivalent promoters should be tested by RT-qPCR. Similarly, the expression level of genes found in GO, KEGG, Reactome, and Wikipathway functional analyses.

Minor concerns:

How do authors explain the high dispersion (in particular in hESCs and NSCs) between the biological replicas of the G4-ChIP-Seq hits (Figure S2 b, c d)?

Line 126 says "Thus, maintenance or acquisition of a promoter G4 favors the transition of bivalent hESC promoters to the active H3K4me3 status in differentiation to CNCCs." Again, the word "maintenance" indicates that the G4 does unfold/re-fold in the transition, which has not been demonstrated. I would use "coincidence" or something similar. The word "favors" suggests that the G4 are responsible for the transition from bivalent to active "H3K4me3". I think authors have to say something like "seems to be related to".

Line 176 "For NSCs, many upregulated genes were associated with". As it is written, "upregulated genes" suggest that these genes were off and turn on; however, it is not so for the NSCs line. I would write "highly expressed in comparison with hESCs".

In line 233, 'differentiation delay' seems more appropriate than 'developmental delay'.

Lines 252-253 state that "G4s are a key component of this regulatory system". However, presented data just suggest a correlation. Further work is needed in this regard.

Similarly, the authors assumed that G4s could provide a rapid and efficient mechanism to switch between silent and active bivalent states on differentiation (lines 258-259). Additional data have to be presented to support this hypothesis.

Reviewer #2 (Remarks to the Author):

In this manuscript, the authors used a ChIP-seq approach against G4 structures present in different human cells to investigate their possible involvement in cell differentiation. Overall, I found this manuscript interesting, although somewhat inflated.

I recommend the authors to answer the following comments to strengthen their results.

1. A main criticism of this manuscript is that conclusions are often overstated in comparison to the nature of the experimental results, because they are based on results of a descriptive nature. For example, the observation that a population of G4s is acquired during differentiation is not the demonstration that "Acquisition of G4s during differentiation is therefore critical to lineage specification" (line 179). I agree that the authors tried to experimentally address a cause-effect relationship with the use of PhenDC3 to stabilize the G4. However, a KO of G4 structures would have been more convincing. However, if the authors moderate their conclusions and correctly address the following comments, I believe that this work can be of wide interest.

2. There is also a large debate about the efficiency and use of anti-G4 antibodies in this field because several laboratories cannot obtain robust results with them. As all the results of this manuscript are essentially based on one antibody, I strongly recommend the authors to provide a very detailed protocol on their use, possibly in the form of a lab protocol mentioning possible caveats, and possible reasons for failures. Actually, when a ChIP assay fails, one usually sees weak enrichment compared with open chromatin. In fact, Input samples are usually weakly enriched over compared with chromatin (see FAIRE-seq). Thus, when peaks overlap extensively with open chromatin, it is important to be sure about the antibody accuracy.

3. Although it looks like that the authors used an "input control" for the ChIP-seq experiments, it is not clear whether they are calling peaks against this input control.

4. The authors found that less than 3% (17,950) of the 700 000 potential G4s are present in hESC. More than half of them were lost upon differentiation, but those present in differentiated cells were also present in the parental hESCs. However, besides their recognition by the antibody, they do not

mention why the antibody recognizes this minor part of all potential G4s. Is it because only these 17,950 potential G4s form G4 structures, or is it because the antibody recognizes only a specific G4 structure? Did the authors analyse in detail the G4 category recognized by the antibody? It would be important to get some insights into this issue.

5. Similarly, the authors found a G4 population specifically associated with the expression of pluripotency genes that are not expressed any more during differentiation (lines 185-187). They should answer the same questions as in point 4.

6. The authors did not check whether the use of PhenDC3 also induces a checkpoint response of the treated cells (CHK1 and CHK2, for example). This is important because it is known that G4 stabilization promotes replication fork arrest and checkpoint responses that might explain their data.

7. Half of bivalent promoters have a G4, and the authors mention that their finding about G4 co-existence with bivalent promoters is the first observation of this kind. However, it was previously reported that G4s at bivalent promoters are associated with DNA replication origins both in mouse and human cells, including with polycomb marks. This should be commented in the manuscript.

8. The observation of nucleosome-depleted regions at G4 was also reported by several previous articles that should be cited.

Minor point:

The use of "for the first time" or equivalent sentences should be avoided (lines 111, 252).

Reviewer #3 (Remarks to the Author):

In the manuscript "G-quadruplex DNA structures in human stem cells and differentiation", Zyner et al investigate G-quadruplex DNA secondary structures (G4s) in human ESCs versus two more restricted stem cell populations, hESC-derived cranial neural crest cells (CNCCs) and neural stem cells (NSCs). They performed G4-ChIP-Seq in the three cell lines and identified a larger number of G4s in hESCs compared to the more lineage-restricted stem cells, with ~80% of the G4s identified in CNCCs and NSCs also found in hESCs. As found in other studies, the majority of these G4s are located in nucleosome depleted regions, and are found in promoters, enhancers, and other regulatory regions. The authors examine G4s in the context of chromatin status, focusing on active (H3K4me3), repressive (H3K27me3), or bivalent domains. They find that ~half of bivalent promoters in hESCs possess a G4, and that maintenance of the G4 in CNCCs is associated with active rather than repressive chromatin state. They additionally compared gene expression changes in the three cell lines, and correlated them with G4 and chromatin status, confirming that G4s are associated with transcriptional activation. Finally, they differentiated hESCs into CNCCs in the presence of a small molecule that stabilizes G4s and determined that this hindered differentiation. Overall, this is a careful, detailed characterization of G4s across these three stem cell types. Although the association of G4s with active transcription, as well as their location in promoters, enhancers, and nucleosome-depleted regions is not novel, the association with bivalent domains and the regulation of differentiation of ESCs provides new information on the importance of G4s during development that is likely to be of broad interest. However, there are several concerns regarding the organization of the manuscript and the conclusions drawn that should be addressed.

1. The manuscript would benefit significantly from a reorganization and the inclusion of key data and discussion in the main paper. In its present form, the introduction is missing key information needed to provide context and many of the important results and discussion points are contained in the supplemental materials. In particular, because many of the findings are dependent on the lineage

relationships between hESCs and CNCCs and NSCs, further detail on their generation and characterization should be included in the primary paper. This would be far more informative than the current phase-contrast images in Fig. 1b that are very difficult to see.

2. One primary conclusion drawn is that G4 "inheritance" maintains the transcription of key stem cell homeostasis genes (e.g. Discussion lines 28-29; Figure 3 title). However, the GO terms identified (Fig. 3) do not appear to be associated with stem cell homeostasis but rather general cellular function (e.g. mRNA processing). Are these G4s and associated transcripts specific to hESCs and CNCCs (or other stem cells), or would the same be found when comparing terminally differentiated cells? The authors should either demonstrate that these G4 inherited transcriptional targets are specific to stem cells, or they should clarify these conclusions.

3. Details on quantification of percent cells with OCT4 expression in Figure 4b are lacking. It is not clear if this represents a single experiment or if biological replicates were performed. Why different numbers of cells were analyzed at different days should also be justified.

G-quadruplex DNA structures in human stem cells and differentiation

Katherine G. Zyner^{1,4}, Angela Simeone^{1,4}, Sean M. Flynn¹, Colm Doyle¹, Giovanni Marsico¹, Santosh Adhikari², Guillem Portella², David Tannahill¹ and Shankar Balasubramanian^{1,2,3},

Affiliations:

¹Cancer Research UK Cambridge Institute, Li Ka Shing Centre, Robinson Way, Cambridge, CB2 0RE, United Kingdom.

²Department of Chemistry, University of Cambridge, Lensfield Road, Cambridge CB2 1EW, United Kingdom.

³School of Clinical Medicine, University of Cambridge, Cambridge, CB2 0SP, United Kingdom.

⁴These authors contributed equally to this work.

*Correspondence to: sb10031@cam.ac.uk

Reviewer #1 (Remarks to the Author):

Major concerns:

1. Throughout the work, the authors have used the words "development" and "differentiation" interchangeably. In this work, it is more appropriate to refer to the ESC-NSC and ESC-CNCC processes as cell differentiation processes. On the other hand, although ESC-NSC and ESC-CNCC differentiation take place during early development, it is not correct to state that experiments done in cultured cells represent 'early developmental' events.

Differentiation is a stepwise process whereby cells progressively acquire their identities during embryonic development directed *in vitro* differentiation has become a widely used *in vitro* model for this (e.g reviewed in Dvash *et al* 2006). We have clarified our wording throughout the manuscript at the referee's request. Some specific examples include:

Line 19: "Here we show that G4s are key genomic structural features linked to *cellular differentiation*."

Line 72: "To gain insights into the role of G4s during development, we utilised a human embryonic stem cell (hESC) system which enables differentiation to be studied *in vitro*, and is an accepted model that recapitulates many aspects of cell lineage specification during human embryogenesis²⁵."

Line 110: "Controlled differentiation provides the opportunity to evaluate whether G4s and chromatin structure are tightly coupled during *cell state transitions*."

2. The main problem of the work is the use cNCC and NSC as cell lines showing different lineage potential. NSCs do not derive from cNCC. The opposite could be possible; i.e., some NSC could derive in NCC (but not cranial). So, NSCs could be even less differentiated than cNCC. Figure 1 has to be changed to solve this problem.

We do not claim that NSCs are derived from CNCCs. To avoid any confusion we have revised the text. We have also emphasised the differing stem cell potential for each daughter cell, both in the revised text (see below) and with a new schematic to replace the original Fig 1b.

Line 75: "hESCs were differentiated into two well-characterised, multipotent stem cells each with differing lineage potentials: cranial neural crest cells (CNCCs)²⁶ and neural stem cells (NSCs²⁷, Fig. 1b-c). NSCs undergo self-renewal and can generate neurons, astrocytes, and oligodendrocytes of the central nervous system²⁸. CNCCs also self-renew but have a broader lineage potential giving rise to cranial neurons and glia as well craniofacial mesodermal derivatives, such as bone, cartilage and smooth muscle^{29,30}."

Fig. 1: G4 abundance is linked to degree of stem cell plasticity...b, Overall study design and schematic showing differing lineage potential of stem cell types generated and analysed. Data generated in this study are indicated in blue text, with published datasets indicated in grey text.

3. Experiments in which ESCs are induced to CNCC (as was done in some experiments in this work), and then those CNCCs are induced to any of the potential derivatives of CNCC (e.g. chondrocytes) could be more appropriate to compare cell differentiation stages.

This is an interesting suggestion for future work that goes beyond the scope of the current manuscript.

4. The word "inheritance" indicates that the G4s are "conserved" as DNA structure during cell differentiation. What is the experimental evidence to sustain that the G4s do not unfold and refold during the differentiation process? In a similar fashion, the word "maintenance" was used in several phases (e.g., line 167), but experimental data supporting this suggestion are scant. In my opinion, it is not correct to use the words inheritance/transmission/propagation since these words suggest that G4 cannot unfold/refold during cell differentiation

We recognise that G4s can undergo dynamic changes between folded and unfolded states (reviewed in Lane *et al* 2008, *Nucleic Acids Research*). Our choice of wording was not intended to suggest otherwise, but was used in the context of defining which G4s are consistently detected, or not, under each condition. We have provided additional text to clarify what we mean **on line 167**: "Hereafter we refer to "G4 maintenance" to describe when promoter G4s are present in both hESCs and the differentiated daughter cell." We also have removed all incidences of the word "inheritance", "conserved", "transmission" and "propagation" so as not to infer anything about G4 dynamics.

Text changes were also made to **Line 103** : "Therefore, while G4s are lost during differentiation, G4s present in differentiated daughter cells largely occur at genomic locations where G4s were already present in the hESC state"

5. A direct check (validation) would be lacking for characteristic genes of each cell type analyzed, both in cases in which the G4 status is maintained (G4E- G4D- or G4E + G4D +) and in those that change (G4E- G4D + or G4E + G4D-) and the corresponding expression levels.

G4-ChIP-qPCR and qRT-PCR measurements have now been added to the manuscript to validate the promoter G4 state and gene expression of a selection of typical genes for each stem cell type (See new Supplementary Fig. 15 – also shown below). Given that G4s are found in highly repetitive genomic regions, specific and robust ChIP-qPCR primer design is challenging. Thus, we focused our validation analysis on G4 promoter regions where robust G4-ChIP-qPCR was most practical.

Supplementary Figure 15: RT-qPCR and G4-ChIP qPCR validation of selected genes uncovered by gene enrichment analyses. a-e, Relative gene expression levels (normalised to GAPDH and relative to hESCs) and f-o, G4 ChIP-qPCR of a selection of genes shown in Fig 4e; Fig 5 b,d,f; Supplementary Fig. 16. The G4-negative *TMCC1* regulatory region⁷¹ was used as internal reference to normalize G4-ChIP qPCR promoter enrichments. Mean +/- stdev, n = 3 biological replicates (see methods). Unpaired t-test: *: $p < 0.05$; **: $p < 0.001$, ***: $p < 0.0001$

6. Does the stabilization of G4 by PhenDC3, which apparently delays the stem feature, increase the levels of H3K4Me3?

Good suggestion. We have checked and were unable to see any differences in H3K4me3 levels after PhenDC3 treatment during hESC differentiation into CNCCs. We present the data below for the referee, but have not included it in the updated manuscript.

Treatment	Biological Replicate	y min	lower	Normalised H3K4me3 (Median)	upper	y max	Number of Cells
Secondary Only	Rep1	0.73	0.79	0.81	0.83	0.89	62587
DMSO	Rep1	2.42	4.62	5.35	6.09	8.29	63647
DMSO	Rep2	1.95	4.26	5.04	5.80	8.11	66516
2 µM PhenDC3	Rep1	0.59	3.70	4.70	5.77	8.87	36592
2 µM PhenDC3	Rep2	1.46	4.37	5.36	6.32	9.23	47140
2 µM PhenDC3	Rep3	1.41	4.26	5.21	6.16	9.02	47206
1 µM PhenDC3	Rep1	2.50	4.75	5.51	6.25	8.50	56588
1 µM PhenDC3	Rep2	2.51	4.73	5.47	6.20	8.42	58169
1 µM PhenDC3	Rep3	2.00	4.43	5.25	6.05	8.48	59928
hESC Secondary Only	Rep1	0.70	0.75	0.76	0.78	0.83	34008
hESC	Rep1	0.89	3.80	4.71	5.74	8.66	20593
hESC	Rep2	1.18	4.04	4.95	5.95	8.81	28489
hESC	Rep3	1.27	4.02	4.88	5.86	8.61	33921

Rebuttal Fig 1: PhenDC3 treatment during CNCC differentiation, does not change H3K4me3 levels. hESCs were differentiated to CNCCs in the presence of PhenDC3 (1 µM and 2 µM) or 0.2% DMSO on µ-Plate 24-well (ibidi). hESC grown in mTESR1 were used as a pluripotency control. At Day 5, cells were fixed permeabilised and stained with primary antibodies against H3K4me3 (anti-rabbit) and Histone H3 (anti-mouse) and detected with donkey anti-rabbit594 and anti-mouse488 secondary antibodies. Normalised H3K4me3 level was calculated as sum H3K4me3 signal/sum of Histone H3 per nucleus (detected with DAPI counterstain). Box and whisker plot shows median Secondary only = wells stained with secondary fluorescent antibody only. BioRep = biological replicate (independent differentiations).

7. Some other drug should be used in similar experiments, even if it damages the DNA a little. It is important to rule out unspecific PhenD3 effects. Alternatively, the effect of G4-helicase depletion could be tested. Hypothetically, G4-helicase depletion might lead to similar results. Even more, the overexpression of G4-helicases should promote cell differentiation.

We have now performed the 5-day CNCC differentiation experiment in the presence of two additional G4-specific small molecules: *N*-methyl mesoporphyrin IX (NMM) and 12459. Both these compounds were tolerated at similar concentrations compared to PhenDC3, and showed low/minimal DNA damage activation (See Supplementary Figure 19). In a concentration-dependent manner, treatment with each of NMM and 12459 resulted in a significant ($p < 0.05$) increase in the proportion of OCT4 expressing cells (at day 3 and day 5) and a decrease in the proportion of SOX10 expressing cells by day 5, albeit at a lower magnitude when compared to PhenDC3. We thank the reviewer for this suggestion, as this new experiment and additional data strengthens our overall conclusion.

Line 282: “Indeed delayed differentiation was also observed when using each of two structurally distinct G4-selective stabilising molecules: *N*-methyl mesoporphyrin IX (NMM)^{61,62} and 12459⁶³ (Supplementary Fig 19). This suggests it is G4 stabilisation, rather than non-specific effects by a given molecule, that are the cause of delayed differentiation.”

Supplementary Figure 19: Treatment with further G4-specific small molecules results in hESC differentiation delay. hESCs were differentiated into CNCCs as described in Figure 6. Samples were taken at the day 3 and day 5 to determine the percentage of cells positive for the pluripotency marker OCT4 or SOX10 (lineage marker for CNCC) by immunofluorescence (IF). **a**, Chemical structures of G4-specific ligands *N*-methyl mesoporphyrin IX (NMM)^{60,61} and 12459⁶². **b**, Representative confocal IF images of differentiating hESCs treated with either 0.02% DMSO, 2 μM

PhenDC3, 2 μ M NMM or 2 μ M 12459. SOX10 = red, OCT4 = green and DAPI (nuclear stain) = blue. Scale bar = 100 μ m. **c-d**, Proportion of (c) OCT4, (d) SOX10 and (e) 53BP1 (DNA damage marker⁹³) positive cells determined in IF studies. n = 7,000 to 75,000 cells obtained from 52 fields of view per sample. Mean \pm sem shown for 3 independent biological replicates *: p < 0.05, **: p < 1E-10, ***: p < 1E-50, ****: p < 1E-100; one-sided Pearson's χ^2 test for proportions (see Supplementary Table 5).

8. The expression of some of the genes having bivalent promoters should be tested by RT-qPCR.

We have now added RT-qPCR measurements for a selection of hESC genes (where expression could be robustly detected by qPCR) with G4 positive and G4 negative bivalent promoters (see Supplementary Figure 9c). Bivalent genes are generally very lowly expressed making them difficult to accurately quantitate using RT-qPCR. As a result, we restricted our analyses to cases in which gene expression was high enough in at least one of the stem cell types to ensure adequate PCR primer efficiency (i.e. 100% \pm 10% primer efficiency (Bustin and Huget 2017, Biomolecular Detection and Quantification)).

Supplementary Fig. 9... c, Relative expression levels (normalised to *GAPDH*) of hESC genes with bivalent promoters that are G4 positive (green) or G4 negative (dark blue). Mean \pm SD of n = 3. *: p < 0.05, unpaired t-tests, See Supplementary Table 4).

9. Similarly, the expression level of genes found in GO, KEGG, Reactome, and Wikipathway functional analyses.

We have now added RT-qPCR measurements to validate the expression of genes found in some of the significant (FDR < 0.005) GO, KEGG, Reactome and Wikipathway functional analyses (see new Supplementary Fig. 15). A summary of the functional gene enrichment terms/categories for each associated gene (top 5 terms per category) is listed in the table below:

G4 status	Symbol	Ensembl ID	GO:BP	KEGG	Reactome	Wikipathway
G4 _E +G4 _D ⁺	PHF23	ENSG00000040633	macroautophagy, protein-containing complex disassembly, regulation of autophagy, regulation of protein modification by small protein conjugation or removal, regulation of macroautophagy			

G4 _E +G4 _D ⁺	KANSL2	ENSG00000139620	covalent chromatin modification, histone modification, peptidyl-lysine modification, protein acylation, protein acetylation		Chromatin modifying enzymes, Chromatin organization, HATs acetylate histones	
G4 _E -G4 _D ⁻	HES7	ENSG00000179111	regulation of cell development, regulation of nervous system development, regulation of neurogenesis, regionalization, pattern specification process	Human papillomavirus infection		Mesodermal Commitment Pathway
G4 _E -G4 _D ⁻	NNAT	ENSG00000053438	regulation of secretion by cell, signal release, regulation of hormone levels, peptide transport, amide transport			
G4 _E +G4 _D ⁻	TERT	ENSG00000164362	Wnt signaling pathway, cell-cell signaling by wnt, anatomical structure homeostasis, positive regulation of cellular protein localization, response to hypoxia,	Human papillomavirus infection, Gastric cancer, Hepatocellular carcinoma, Human T-cell leukemia virus 1 infection	Signaling by WNT, TCF dependent signaling in response to WNT,	Head and Neck Squamous Cell Carcinoma, Lung fibrosis
G4 _E +G4 _D ⁻	EPHA1	ENSG00000040633				
G4 _E +G4 _D ⁻	PRDM14	ENSG00000147596	histone methylation, histone modification, covalent chromatin modification, protein alkylation, protein methylation		Transcriptional regulation of pluripotent stem cells	Endoderm Differentiation
G4 _E -G4 _D ⁺	TWIST1	ENSG00000122691	skeletal system development, mesenchymal cell differentiation, skeletal system morphogenesis, embryonic skeletal system development, mesenchyme development	Proteoglycans in cancer	Interleukin-4 and Interleukin-13 signaling, Signaling by Interleukins	Epithelial to mesenchymal transition in colorectal cancer, Neural Crest Differentiation, Transcription factor regulation in adipogenesis, Adipogenesis
G4 _E -G4 _D ⁺	TWIST2	ENSG00000233608	osteoblast differentiation, ossification, regulation of osteoblast differentiation	Proteoglycans in cancer		Epithelial to mesenchymal transition in colorectal cancer
G4 _E -G4 _D ⁺	RFX4	ENSG00000111783	spinal cord development, regionalization, pattern specification process, negative regulation of smoothed signaling pathway, dorsal/ventral pattern formation			
G4 _E -G4 _D ⁺	DLL1	ENSG00000198719	gliogenesis, regulation of neurogenesis, regulation of nervous system development, spinal cord development, regulation of cell development	Breast cancer, Notch signaling pathway, Endocrine resistance,	Signaling by NOTCH2, Signaling by NOTCH1 in Cancer, Signaling by NOTCH1 PEST Domain Mutants in Cancer, Signaling by NOTCH1 HD+PEST Domain Mutants in Cancer, Constitutive Signaling by NOTCH1 HD+PEST Domain Mutants	Neural Crest Differentiation, Breast cancer pathway, Canonical and Non-canonical Notch signaling, Notch Signaling, Notch Signaling Pathway Netpath
G4 _E -G4 _D ⁺	ING4	ENSG00000111653				

G4 _E -G4 _D ⁺	SOX11	ENSG00000176887	skeletal system development, mesenchymal cell differentiation, skeletal system morphogenesis, embryonic skeletal system development, mesenchyme development			
G4 _E -G4 _D ⁺	NR2F1	ENSG00000175745			Nuclear Receptor transcription pathway	Adipogenesis

Minor concerns:

10. How do authors explain the high dispersion (in particular in hESCs and NSCs) between the biological replicas of the G4-ChIP-Seq hits (Figure S2 b, c d)?

Generally, variability in peak numbers called across biological replicates is very common in ChIP-seq, and thus ENCODE requires a minimum of two biological replicates for acceptance (Landt *et al* 2012, Genome Research). Variability in ChIP-seq libraries can be partially attributed to confounders such as cell confluency, age, and passage number, as well as inevitable differences in the technical processing of samples such as sonication, ChIP, library preparation and sequencing. We do not feel that there is unusually high dispersion between our biological replicates and our data is in line with what we have previously observed for G4-ChIP-seq (see Supplementary Fig. 1d from Spiegel *et al* 2021, *Genome Research*; and Supplementary Fig. 1a from Hansel-Hertsch *et al* 2016, *Nature Genetics* and our G4-ChIP-seq methods paper: Hansel-Hersch *et al* 2018, *Nature Methods*).

In the current study, we generated a total of 9 ChIP-seq libraries per cell line (3 technical replicates for each of the 3 biological replicates) and only used the G4 consensus regions (defined as regions observed in 2 out of 3 technical replicates in 2 out of 3 biological replicates) for all downstream analyses. Furthermore, as described in the method section ‘G4-ChIP-Seq and library preparation’:
Line 617: “*Each biological replicate for hESCs, CNCCs and NSCs was sonicated, ChIP’d, library prepped and sequenced together, and additional biological replicates performed on a different day*” to minimise as many of the previously listed confounding variables as possible.

11. **Line 126** says "Thus, maintenance or acquisition of a promoter G4 favors the transition of bivalent hESC promoters to the active H3K4me3 status in differentiation to CNCCs." Again, the word "maintenance" indicates that the G4 does unfold/re-fold in the transition, which has not been demonstrated. I would use "coincidence" or something similar. The word "favors" suggests that the G4 are responsible for the transition from bivalent to active "H3K4me3". I think authors have to say something like "seems to be related to".

We have changed the word “favours” to “appears to be related to” on line 126 (now line 175): “*Thus, maintenance or acquisition of a promoter G4 appears to be related to the transition of bivalent hESC promoters towards an active H3K4me3 state in CNCC differentiation.*” As stated above (point 4) above, we have explicitly defined our use of the word “maintenance”: on line 167: “*Hereafter we refer to “G4 maintenance” to describe when promoter G4s are present in both hESCs and the differentiated daughter cell.*”

12. **Line 176:** "For NSCs, many upregulated genes were associated with". As it is written, "upregulated genes" suggest that these genes were off and turn on; however, it is not so for the NSCs line. I would write "highly expressed in comparison with hESCs".

We have amended the relevant text (now line 238): "*Likewise for NSCs, many genes with increased expression that acquire a G4 were related to with neurodevelopmental pathways ...*"

13. **In line 233,** 'differentiation delay' seems more appropriate than 'developmental delay'.

We have amended the text accordingly (now line 304): "*This resulted in a differentiation delay due to failure of pluripotency exit and suggests that the high abundance of G4s in hESC relative to CNCCs and NCs acts to maintain the pluripotent state.*"

14. **Lines 252-253** state that "G4s are a key component of this regulatory system". However, presented data just suggest a correlation. Further work is needed in this regard. Similarly, the authors assumed that G4s could provide a rapid and efficient mechanism to switch between silent and active bivalent states on differentiation (**lines 258-259**). Additional data have to be presented to support this hypothesis.

We have moderated our wording to clarify that at this stage any role of G4s in providing a molecular switch is an idea to be tested in future work:

Line 252-253 (now Lines 325-327): "*Histone modifications are dynamic features important in the control of gene expression and differentiation⁶⁶ and here we provide evidence that G4 structures may act as an additional layer within an epigenetic regulatory system.*"

Lines 258-259 (now Lines 331-341): "*Our findings may extend mathematical modelling that predicts chromatin bivalency at CpGI exists as a "bistable" system, frequently switching between active and silent chromatin states⁶⁷. For instance, G4 folding/unfolding could theoretically provide a rapid and efficient mechanism to modulate between active and silent bivalent states. Indeed, G4s have been identified as binding sites for effector proteins with histone-modifying activities (e.g. ATRX and LSD1⁹)... Further work will be necessary to understand the detailed mechanistic interplay between the G4 and histone landscapes and subsequent transcriptomic changes.*"

Reviewer #2 (Remarks to the Author):

1. A main criticism of this manuscript is that conclusions are often overstated in comparison to the nature of the experimental results, because they are based on results of a descriptive nature. For example, the observation that a population of G4s is acquired during differentiation is not the demonstration that “Acquisition of G4s during differentiation is therefore critical to lineage specification” (line 179). I agree that the authors tried to experimentally address a cause-effect relationship with the use of PhenDC3 to stabilize the G4. However, a KO of G4 structures would have been more convincing. However, if the authors moderate their conclusions and correctly address the following comments, I believe that this work can be of wide interest.

CRISPR knockout of a G4 structure is an important approach to gain further insight into the biological roles of G4 at a gene promoter. However, this experimental approach can be confounded by any changes to the G4-motif sequence in a promoter also disrupting the canonical binding sites of transcription factors such as SP1 (Spiegel *et al* 2021, *Genome Biology* and Lago *et al* 2021, *Nature Communications*). Thus, this makes it difficult to definitively attribute any observed effects solely due to loss of a G4 structure. We acknowledge that most of our study is based on genomic correlations and hence we have moderated the conclusions of the manuscript accordingly. Examples include:

Line 179 (now Line 243) “*Acquisition of G4s during differentiation may therefore play a role in lineage specification*”.

Line 285-288: “*High G4 abundance in hESCs appears to be associated with the pluripotent state, whereas dynamic changes in the G4 landscape are coupled to the transcriptional reprogramming that takes place during differentiation.*”

Line 312-315: “*We propose, therefore, that promoter G4s that are ‘maintained’ from hESCs to daughter cells upon differentiation sustain expression of associated genes with less transcriptional variability.*”

2a. There is also a large debate about the efficiency and use of anti-G4 antibodies in this field because several laboratories cannot obtain robust results with them. As all the results of this manuscript are essentially based on one antibody, I strongly recommend the authors to provide a very detailed protocol on their use, possibly in the form of a lab protocol mentioning possible caveats, and possible reasons for failures.

Anecdotally, we have heard that some labs have been unable to perform G4-ChIP-seq satisfactorily. The reasons for this are unclear. In our hands it is important to use BG4 as a scFv FLAG-tagged antibody as originally described in our *Nature Chemistry* publication (Biffi *et al*, 2013). We suspect that some labs are using the IgG reformatted version of BG4, which we have not validated. We also note that other labs have independently had success with G4 ChIP (Richter Lab: Lago *et al* 2021, *Nature Communications*; Bhakat Lab: Roychoudhury *et al* 2020, *PNAS*, Wang Lab: Li *et al* 2020, *Nature Chemical Biology*; Liang Lab: Li *et al* 2021, *Genome Research*). We previously published a detailed description of G4-ChIP-seq in *Nature Protocols* (Hansel-Hertsch *et al* 2018). Updates to that protocol are described in our subsequent papers in (e.g. Hansel-Hertsch *et al* 2020, *Nature Genetics*; Spiegel *et al* 2021, *Genome Biology*; Shen and Varshney *et al* 2021, *Genome Biology*). Furthermore, for the human stem cells described in this manuscript, we have gone to some effort to describe any relevant changes in full (see **Line 579**, methods section: *G4-ChIP-Seq and library preparation*).

Given these points and that the current manuscript is not a methods paper, we do not think that it is appropriate to rewrite more or less the same detailed protocol as previously published. To assist users in the field further we will publish our current best protocol on our website and keep it updated with any future improvements: <https://www.balasubramanian.co.uk/research-resources>.

2b. Actually, when a ChIP assay fails, one usually sees weak enrichment compared with open chromatin. In fact, input samples are usually weakly enriched over compared with chromatin (see FAIRE-seq). Thus, when peaks overlap extensively with open chromatin, it is important to be sure about the antibody accuracy.

We and others have extensively characterised the BG4 antibody, showing that it has very high selectivity and low nM affinity for G4 structures (see Biffi *et al* 2013, *Nature Chemistry*; King *et al*, 2021, *JACS*; Jabadekar *et al* 2020, *DNA Research*). G4s maps identified using BG4 antibody show similar features as maps generated using independent G4 antibodies (Liu *et al* 2016, *Cell Chemical Biology*), G4 probes (Zheng *et al* 2020, *NAR*) while independent mapping approaches (Kouzine *et al* 2017 *Cell Systems*), show strong enrichment of G4s at open chromatin promoters of highly transcribed genes. The loss of peaks during differentiation is unlikely to be due to failure of the G4-ChIP-seq assay. For each chromatin sample, we always confirm enrichment by G4-ChIP-qPCR for positive control peaks and all our experiments have extensive technical and biological replicates (see answer to referee 1, point 10).

3. Although it looks like that the authors used an "input control" for the ChIP-seq experiments, it is not clear whether they are calling peaks against this input control.

To be clear, yes, we do use the input to call peaks using MACS2, and this is stated in our methods section (See section: *Sequencing data processing, G4 and gene-expression differential analysis and G4-ChIP-Seq and library preparation*). There is a specific input control for each biological replicate (total of 9 inputs, 3 per cell line). To prevent any further confusion, we have added an input track to the genome browser tracks in Figure 1e (see also below), when the G4-ChIP-seq tracks are first shown in the manuscript and have added the following sentences to the methods section:

Line 596: "For each biological replicate, an input sample was processed in parallel with omission of the BG4 antibody immunoprecipitation step."

Line 672: "For G4-ChIP-seq, local enrichments for each ChIP were called from an input control library (one separate input control for each biological replicate)."

Fig. 1: G4 abundance is linked to degree of stem cell plasticity... e, Genome browser view of G4 signal for all three stem cell types across the promoters of the genes *FZD2*, *MSX1*, *PTPRZ1* and *CHD1*. Yellow box highlights regions where G4s overlaps open chromatin (defined by ATAC-seq; ATAC) and genome sites which have the ability to fold into G4 structures *in vitro* (called OQs, observed quadruplex sequences¹³). Genomic coordinates indicated at top.

4. The authors found that less than 3% (17,950) of the 700 000 potential G4s are present in hESC. More than half of them were lost upon differentiation, but those present in differentiated cells were also present in the parental hESCs. However, besides their recognition by the antibody, they do not mention why the antibody recognizes this minor part of all potential G4s. Is it because only these 17,950 potential G4s form G4 structures, or is it because the antibody recognizes only a specific G4 structure? Did the authors analyse in detail the G4 category recognized by the antibody? It would be important to get some insights into this issue.

This complex issue was discussed in some detail in our first G4-ChIP-seq manuscript (Hansel-Hertsch *et al Nature Genetics* 2016). Given that different cell types display different G4s detected by G4-ChIP-seq (also recently shown independently by another lab: Lago *et al* 2021, *Nature Communications*), the simplest explanation is that some G4s are favoured to fold in one cell type but are unfolded in another. We and others have suggested that this may be in part due to the nature of the local chromatin environment (see: Shen and Varshney *et al* 2020, Roychoudhury *et al* 2020, *PNAS*). Furthermore, deletion of particular G4-interacting proteins has also been demonstrated to modulate the G4 landscape in immunofluorescence (e.g. CST complex (Zhang *et al* 2019, *NAR*), ATRX (Wang *et al* 2019, *Nature Communications*), HERC2, BLM, WRN and RPA (Wu *et al* 2018, *Cancer Research*)) and CUT&TAG experiments (e.g. ATRX, see Figure 3 from Teng *et al* 2021, *Nature Communications*).

As G4s have multiple topologies and many different sequences, it is reasonable to ask if the BG4 antibody recognises every possible G4 structure. Our previous computational analysis of the types of G4s detected in G4-ChIP-seq showed that a large range of different G4 types are detected (Hansel-Hertsch *et al* 2016, *Nature Genetics*; Hansel-Hertsch *et al* 2018, *Nature Protocols*) and so we have

not observed any bias that would exclude a particular G4 type. We have included further text on this and added an additional analysis to **Supplementary Figure 2 (h-i)**. See below:

Line 91-93: “*We confirmed that BG4 was indeed recognising a broad spectrum of G4 structural types (Supplementary Fig. 2h-i) as for previous studies^{14,32}.*”

Addition to Supplementary Figure 2:

h) Enrichment analysis for different G4 structural classes. The total number of G4 regions in each structural class, per stem cell line are shown. Loop size 1–3, 4–5 and 6–7, indicates that at least one loop of this length is present in the G4; long loop indicates a G4 with any loop of length >7 (up to 12 for any loop and 21 for the middle loop); simple bulge indicates a G4 with a bulge of 1–7 bases in one G-run or multiple 1-base bulges; 2-tetrads / complex bulge indicates G4s with two G-bases per G-run or several bulges of 1–5 bases; and other indicates other G4-types that do not fall into the former categories. **i)** Fold enrichment for each structural class in (g) compared to random (average of 3 randomisations). Higher enrichment values equates to higher likelihood of being present among the G4 regions

5. Similarly, the authors found a G4 population specifically associated with the expression of pluripotency genes that are not expressed any more during differentiation (lines 185-187). They should answer the same questions as in point 4.

See also response to referee 2, point 4. We performed a similar computational analysis to determine the enrichment of different G4 structural classes in hESC-specific G4 sequences (those not found in either daughter cell). While this was a good suggestion by the reviewer to determine whether a particular sub-class of G4 is associated with the expression of pluripotency genes, it does not appear to be the case. We have not included this in the manuscript and include it for the referee below:

6. The authors did not check whether the use of PhenDC3 also induces a checkpoint response of the treated cells (CHK1 and CHK2, for example). This is important because it is known that G4 stabilization promotes replication fork arrest and checkpoint responses that might explain their data.

We have shown that treatment with concentrations up to 2 μ M of PhenDC3 does not result a growth arrest or increase in DNA damage in hESC (see Supplementary Figure 17a-b). Now, we have performed the following additional experiments as requested. Firstly, an immunofluorescence experiment on day 3 and day 5 differentiating hESCs showed no increase in DNA damage, using the known DNA damage marker 53BP1 (Rappold *et al* 2001, *Journal of Cell Biology*; See Supplementary Figure 19b and e). Secondly, we have also included western blot data to further substantiate that there is also no checkpoint response (e.g. phosphorylation of CHK1 or CHK2) upon PhenDC3 treatment (up to 2 μ M) in differentiating hESCs (see Supplementary Fig. 17c-e).

Supplementary Figure 17: Differentiation delay with PhenDC3 treatment is not due to growth arrest or induction of DNA damage.... c) hESCs were differentiated to CNCCs in the presence of PhenDC3 (1 μ M and 2 μ M) or 0.2% DMSO. On day 5, cell lysates were probed with antibodies against CHK1, phosphorylated CHK1(Ser354), CHK2, phosphorylated CHK2 (Thr68) and GAPDH control by western blotting. **d-e)** Normalised protein levels for d) pCHK1 and e) pCHK2. N = three biological replicates (mean \pm standard deviation). hESCs treated with 100 nM etoposide for 3 hours was used as a positive control.

7. Half of bivalent promoters have a G4, and the authors mention that their finding about G4 co-existence with bivalent promoters is the first observation of this kind. However, it was previously reported that G4s at bivalent promoters are associated with DNA replication origins both in mouse and human cells, including with polycomb marks. This should be commented in the manuscript.

As requested, we have now cited earlier studies which have observed sequence motifs encoding G4s as being associated with bivalent promoter and polycomb marks (see lines below), though those studies did not determine folded G4 structure formation at those sites. A key focus of our work is to detect G4s that physically fold into secondary structures in cellular chromatin. This is an important distinction as not all sequence motifs with potential to fold into a G4 actually fold into a detectable structure in chromatin.

Line 157-162: “Our observation is the first that we are aware of showing that folded G4 structures can physically co-exist with repressive histone marks in the context of bivalency. This provides experimental support for findings from computational studies that predict G4 sequence motif association with bivalent and polycomb-associated chromatin at DNA⁴⁴⁻⁴⁶. Our data thus highlights G4s as prevalent structural features in poised genes involved in developmental decisions.”

8. The observation of nucleosome-depleted regions at G4 was also reported by several previous articles that should be cited.

Our work focuses on detectable, folded G4 structures. It is correct that several publications have shown a correlation of computationally predicted G4 sequence motifs, rather than detected structures, in NDRs. As requested, we cite additional references.

Line 59: “In keeping with computational predictions, endogenous G4s in chromatin are primarily located in nucleosome-depleted regions (NDRs) of highly active promoters^{14,18,19}.”

Minor point:

9. The use of “for the first time” or equivalent sentences should be avoided (lines 111, 252).

Both these lines refer to the observation of detecting G4 at bivalent promoters. As described above in our answer to Point 7 (Line 157-162), we believe we are the first to observe a folded G4 structure at these locations. We have amended the text to reflect this:

Line 252 (now line 300): “In addition to highlighting G4s as a feature associated with active promoters, our work has also revealed folded G4 structures are features of many bivalent promoters.”

Reviewer #3 (Remarks to the Author):

1. The manuscript would benefit significantly from a reorganization and the inclusion of key data and discussion in the main paper. In its present form, the introduction is missing key information needed to provide context and many of the important results and discussion points are contained in the supplemental materials. In particular, because many of the findings are dependent on the lineage relationships between hESCs and CNCCs and NSCs, further detail on their generation and characterization should be included in the primary paper. This would be far more informative than the current phase-contrast images in Fig. 1b that are very difficult to see.

We have added further explanatory text (see Lines 75-85) and new panel Fig. 1b-c (Fig. 1b is also included on page 2 of rebuttal, see below for Fig. 1c) describing the lineage relationship between the stem cells. We have also reorganised the manuscript to improve clarity by including an extended introduction, removing the majority of Supplementary Discussion and integrating it into key parts into the main text and adding two new main figures: Fig. 2: G4s are found in stem cell regulatory elements and Fig. 5: Promoter G4 landscape changes are related to stem cell identity.

Lines 75-85: *“hESCs were differentiated into two well-characterised, multipotent stem cells with differing lineage potentials: cranial neural crest cells (CNCCs)²⁶ and neural stem cells (NSCs)²⁷, Fig. 1b-c). NSCs undergo self-renewal and can generate neurons, astrocytes, and oligodendrocytes of the central nervous system²⁸. CNCCs also self-renew but have a broader lineage potential giving rise to cranial neurons and glia as well craniofacial mesodermal derivatives, such as bone, cartilage and smooth muscle^{29,30}. We first confirmed stem cell derivation and identity using a range of established cell-lineage-specific antibody markers by immunofluorescence microscopy (IF), flow cytometry experiments and RNA-seq (Fig. 1c and Supplementary Fig. 1). For example, hESCs showed specific expression of OCT4 (POU5F1) and NANOG, NSCs specifically expressed PAX6 and SOX1 and CNCCs specifically expressed TFAP2A and TWIST1.”*

Fig. 1: G4 abundance is linked to degree of stem cell plasticity.... c, Immunofluorescent microscopy images for hESC (NANOG, OCT4), CNCC (p75NTR and TFAP2A) and NSC (PAX6 and NESTIN) markers. Inset: cell lineage marker merge with DAPI. Scale bar = 50 μ m

2. One primary conclusion drawn is that G4 “inheritance” maintains the transcription of key stem cell homeostasis genes (e.g. Discussion lines 28-29; Figure 3 title). However, the GO terms identified (Fig. 3) do not appear to be associated with stem cell homeostasis but rather general cellular function (e.g. mRNA processing). Are these G4s and associated transcripts specific to hESCs and CNCCs (or other stem cells), or would the same be found when comparing terminally differentiated cells? The authors should either

demonstrate that these G4 inherited transcriptional targets are specific to stem cells, or they should clarify these conclusions.

Most genes that maintain fundamental cellular processes and homeostatic functions are present in all cell types. We did not mean to imply stem cell-only homeostasis genes, instead we meant genes in stem cells that are involved in basic cell functions and homeostasis. We describe that many of such genes carried a G4 structure. We have amended the text for clarity as requested.

Line 21: “*G4s are prevalent in enhancers and promoters. G4s that are found in common between embryonic and downstream lineages are tightly linked to transcriptional stabilisation of genes involved in essential cellular functions as well as transitions in the histone post-translation modification landscape.*”

Line 220: “*Gene Ontology (GO) enrichment analysis for genes that are not differentially expressed and were G4_E⁺ G4_D⁺ upon differentiation, revealed essential cellular programs as highest-ranking terms ...”*

Line 229: “*Overall, these results show that G4 maintenance maybe an important chromatin feature linked to the transcriptional stabilisation of genes essential for key cellular functions.*”

Line 297: “*During differentiation, we reveal that promoter G4 ‘maintenance’ stabilises transcriptional levels of genes for essential cellular/homeostasis functions and is associated with important developmental changes in the histone modification landscape.*”

Line 307: “*Our work provides novel insights into how the transcription of key lineage specification and essential cellular functions/homeostasis pathways are maintained during differentiation in the face of transcriptional noise⁶⁴.*”

3. Details on quantification of percent cells with OCT4 expression in Figure 4b are lacking. It is not clear if this represents a single experiment or if biological replicates were performed. Why different numbers of cells were analyzed at different days should also be justified.

Fig. 4b (now Supplementary Fig. 17g) quantified the proportion of cells with higher levels of OCT4-EGFP expression compared to DMSO controls in the live cell flow cytometry experiment. This represents one experiment that complements the immunofluorescence (Fig. 6b-c and Supplementary Fig.19) and RNA-seq results (Fig. 6e and Supplementary Fig.18, both have a minimum of three independent biological replicates. Flow cytometry was initially used to determine which timepoints to analyse in further detail. A minimum of 10,000 cells is normally required for flow cytometry experiments, however, on day 1 of the hESC differentiation there was not enough cells in the plate so we instead just took the full sample (5,000 cells).

We have added the relevant information on monitoring OCT4 levels, see below:

Legend of Supplementary Figure 17: **g**, Proportion of cells with higher levels of OCT4-EGFP expression compared to DMSO controls. n = 10,000 cells analysed per day (apart from day 1 where the whole sample was analysed (5,000 cells)). *One biological replicate. See Supplementary Table 4.*

For clarity, we included the number of cells analysed and number of biological replicates for all other experiments in the relevant figure legends.

Rebuttal References

- Biffi, G., Tannahill, D., McCafferty, J. & Balasubramanian, S. Quantitative visualization of DNA G-quadruplex structures in human cells. *Nat. Chem.* **5**, 182–186 (2013).
- Dvash, T., Ben-Yosef, D. & Eiges, R. Human Embryonic Stem Cells as a Powerful Tool for Studying Human Embryogenesis. *Pediatr. Res.* *2006* **602** **60**, 111–117 (2006).
- Hansel-Hertsch, R. et al. G-quadruplex structures mark human regulatory chromatin. *Nat. Genet.* **48**, 1267–1272 (2016).
- Hansel-Hertsch, R., Spiegel, J., Marsico, G., Tannahill, D. & Balasubramanian, S. Genome-wide mapping of endogenous G-quadruplex DNA structures by chromatin immunoprecipitation and high-throughput sequencing. *Nat. Protoc.* **13**, 551–564 (2018).
- Hänsel-Hertsch R, Simeone A, Shea A, Hui WWI, Zyner KG, Marsico G, Rueda OM, Bruna A, Martin A, Zhang X, Adhikari S, Tannahill D, Caldas C, Balasubramanian S. Landscape of G-quadruplex DNA structural regions in breast cancer. *Nat Genet.* 2020;52(9):878–83.
- Javadekar, S. M., Nilavar, N. M., Paranjape, A., Das, K. & Raghavan, S. C. Characterization of G-quadruplex antibody reveals differential specificity for G4 DNA forms. *DNA Res.* **27**, (2020).
- King, J. J. et al. DNA G-Quadruplex and i-Motif Structure Formation Is Interdependent in Human Cells. *J. Am. Chem. Soc.* **142**, 20600–20604 (2020).
- Kouzine F, Wojtowicz D, Baranello L, Yamane A, Nelson S, Resch W, et al. Permanganate/S1 nuclease footprinting reveals non-B DNA structures with regulatory potential across a mammalian genome. *Cell Syst.* 2017;4:344–356.e7
- Lago, S., Nadai, M., Cernilogar, F.M. et al. Promoter G-quadruplexes and transcription factors cooperate to shape the cell type-specific transcriptome. *Nat Commun* **12**, 3885 (2021)
- Lane, A. N., Chaires, J. B., Gray, R. D. & Trent, J. O. Stability and kinetics of G-quadruplex structures. *Nucleic Acids Res.* **36**, 5482–5515 (2008)
- Landt, S. G. et al. ChIP-seq guidelines and practices of the ENCODE and modENCODE consortia. *Genome Res.* **22**, 1813–1831 (2012).
- Li, C. et al. Ligand-induced native G-quadruplex stabilization impairs transcription initiation. *Genome Res.* **31**, 1546–1560 (2021).
- Liu, H. Y. et al. Conformation selective antibody enables genome profiling and leads to discovery of parallel G-quadruplex in human telomeres. *Cell Chem. Biol.* **23**, 1261–1270 (2016).
- Rappold, I., Iwabuchi, K., Date, T. & Chen, J. Tumor Suppressor P53 Binding Protein 1 (53bp1) Is Involved in DNA Damage–Signaling Pathways. *J. Cell Biol.* **153**, 613 (2001).

- Roychoudhury, S. et al. Endogenous oxidized DNA bases and APE1 regulate the formation of G-quadruplex structures in the genome. *Proc. Natl. Acad. Sci. U. S. A.* (2020) doi:10.1073/pnas.1912355117.
- Shen J, Varshney D, Simeone A, et al. Promoter G-quadruplex folding precedes transcription and is controlled by chromatin. *Genome Biol.* 2021;22(1):143. Published 2021 May 7. doi:10.1186/s13059-021-02346-7
- Spiegel, J., Cuesta, S.M., Adhikari, S. *et al.* G-quadruplexes are transcription factor binding hubs in human chromatin. *Genome Biol* **22**, 117 (2021)
- Teng, YC., Sundaresan, A., O'Hara, R. *et al.* ATRX promotes heterochromatin formation to protect cells from G-quadruplex DNA-mediated stress. *Nat Commun* **12**, 3887 (2021)
- W, H. et al. Yin Yang 1 contains G-quadruplex structures in its promoter and 5'-UTR and its expression is modulated by G4 resolvase 1. *Nucleic Acids Res.* 40, 1033–1049 (2012).
- Wang, Y. et al. G-quadruplex DNA drives genomic instability and represents a targetable molecular abnormality in ATRX-deficient malignant glioma. *Nat. Commun.* **10**, 943 (2019).
- Wu, Y., Shin-ya, K. & Brosh, R. M. Jr FANCD1 helicase defective in Fanconi anemia and breast cancer unwinds G-quadruplex DNA to defend genomic stability. *Mol. Cell Biol.* **28**, 4116–4128 (2008).
- Zhang, M. et al. Mammalian CST averts replication failure by preventing G-quadruplex accumulation. *Nucleic Acids Res.* **47**, 5243–5259 (2019).
- Zheng K, Zhang J, He Y, Gong J, Wen C, Chen J, Hao YH, Zhao Y, Tan Z. Detection of genomic G-quadruplexes in living cells using a small artificial protein. *Nucleic Acids Res.* 2020;48(20):11706–20.

REVIEWERS' COMMENTS

Reviewer #1 (Remarks to the Author):

I recommend acceptance of this new version of the manuscript. The authors have presented very important additional experimental data and better contextualized the biological problem. Altogether, these changes have driven significant improvement of the work.

Reviewer #2 (Remarks to the Author):

The authors have correctly answered to my suggestions to improve the manuscript. I do not have further remarks for this interesting manuscript.

Reviewer #3 (Remarks to the Author):

The revised manuscript "G-quadruplex DNA structures in human stem cells and differentiation" contains substantial additional control data that enhance the rigor and clarify a number of the interpretations. Reorganization of the main text and clarification of terminology and conclusions further strengthen the manuscript. With these changes, the authors have addressed my primary concerns and I recommend publication of this improved manuscript that provides new information on the importance of G4s during development, which is likely to be of broad interest.